



# Optimizing a backscatter forward operator using Sentinel-1 data over irrigated land

Sara Modanesi[1,2,4], Christian Massari[1], Alexander Gruber[2], Hans Lievens[2], Angelica Tarpanelli[1], Renato Morbidelli[3], Gabrielle J. M. De Lannoy[2]

[1]Research Institute for Geo-hydrological Protection, National Research Council, Via della Madonna Alta 126, 06128 Perugia, Italy
[2]Department of Earth and Environmental Sciences, KU Leuven, Heverlee, Belgium
[3]Dept. of Civil and Environmental Engineering of Perugia, Via G. Duranti 93, 06125 Perugia, Italy
[4]DICEA Dept. of Civil and Environmental Engineering, University of Florence, Via di S. Marta 3, 50139 Firenze, Italy

Correspondence to: Christian Massari (christian.massari@irpi.cnr.it)

**Abstract.** Worldwide, the amount of water used for agricultural purposes is rising and the quantification of irrigation is becoming a crucial topic. Because of the the limited availability of in situ observations, an increasing number of studies is focusing on the synergistic use of models and satellite data to detect and quantify irrigation. The parameterization of irrigation in large scale Land Surface Models (LSM) is improving, but it is still hampered by the lack of information about dynamic crop rotations or the extent of irrigated areas, and the mostly unknown timing and amount of irrigation. On the other hand, remote sensing observations offer an opportunity to fill this gap as they are directly affected by, and hence potentially able to detect, irrigation. Therefore, combining LSMs and satellite information through data assimilation can offer the optimal way to quantify the water used for irrigation.

The aim of this study is to optimize a land modelling system, consisting of the Noah-MP LSM, coupled with a backscatter observation operator, over irrigated land in order to simulate backscatter predictions. This is a first step towards building a reliable data assimilation system to ingest level-1 Sentinel-1 observations. In this context, we tested how well modelled soil moisture and vegetation estimates from the Noah-MP LSM running within the NASA Land Information System (LIS), with or without irrigation simulation, are able to capture the signal of high-resolution Sentinel-1 backscatter observations over the Po river Valley, an important agricultural area in Northern Italy. Next, aggregated 1-km Sentinel-1 backscatter observations were used to calibrate a Water Cloud Model (WCM) as observation operator using simulated soil moisture and Leaf Area Index estimates. The WCM was calibrated with and without activating an irrigation scheme in Noah-MP and considering two different cost functions. Results demonstrate that activating an irrigation scheme provides the optimal calibration of the WCM, even if the irrigation estimates are inaccurate. The Bayesian optimization is shown to result in the best unbiased calibrated system, with minimal chance of having error cross correlations between the model and observations. Our time series analysis further confirms that Sentinel-1 is able to track the impact of human activities on the water cycle, highlighting its potential to improve irrigation, soil moisture and vegetation estimates via future data assimilation.


# 1 INTRODUCTION

Over the last century, the global water withdrawal grew 1.7 times faster than the population (FAO, 2006). This aggravates the concern over the sustainability of water use as demand for agricultural uses continues to increase (Foley et al., 2011; FAO AQUASTAT http://www.fao.org/nr/water/aquastat/water_use/index.stm, last access 20 May 2021). The strong impact of irrigation on the global water budget is highlighted by many studies and it has been estimated that about 87% of the global fresh water withdrawals have been used for agriculture (Douglas et al., 2009). Accordingly, the quantification of irrigation on a

regional to global scale has become a hot research topic.

Correctly quantifying irrigation in Earth system models can serve two purposes. On the one hand, it can help improve water management (Le Page et al., 2020, Bretreger et al., 2020), on the other hand, it allows to quantitatively assess its effects on the terrestrial water, carbon and energy cycles (Haddeland et al., 2007; Breña-Naranjo et al., 2014; Hu et al., 2016; Qian et al. 2020). Indeed, results of large-scale irrigation studies using land surface models (LSMs) have demonstrated that irrigation increases

soil moisture and evapotranspiration (ET), and consequently latent heat flux with a decrease in sensible heat flux (i.e., Badger & Dirmeyer, 2015; Lawston et al., 2015; Ozdogan et al., 2010b).

Despite the significant impact of irrigation on the water and energy cycles, its simulation within LSMs is not yet common practice (Girotto et al., 2017). Attempts to simulate irrigation in LSMs have in the past relied on different parameterizations of well-known irrigation systems (like sprinkler, flood, and drip systems; Ozdogan et al. 2010b; Evans and Zaitchik, 2008) either

without specifying the source of water withdrawals and by relying on additional fictitious rainfall (Ozdogan et al. 2010b) or taking irrigation water from groundwater (Nie et al., 2018). Irrigation is normally applied when soil moisture drops below a user-defined threshold (Ozdogan et al. 2010b), typically dependent on the soil properties obtained via soil texture maps.

Moreover, LSMs equipped with irrigation schemes need to be provided with auxiliary information about crop types and whether or not the crops are irrigated. This is because different crop types are characterized by different rooting depths, which means

they require more or less water to restore root zone field capacity. This information is normally gathered from static maps derived from statistical analysis and/or remote sensing (Ozdogan et al., 2010b; Monfreda et al., 2008; Salmon et al., 2015) collected during specific historical periods which are normally different from to the desired period of analysis. It is thus clear that the modelling of irrigation is subject to many simplifying assumptions, which span from neglecting the year-to-year crop variability and the irrigation system used to the definition of irrigation application times based on water availability and crop conditions

rather than actual farmer decisions.

Remote sensing (RS) technologies offer the opportunity to observe directly the Earth surface and its changes, and hence are potentially able to monitor irrigated lands worldwide (Ambika et al., 2016; Gao et al., 2018; Bousbih et al., 2018; Bazzi et al., 2019; Le Page et al., 2020; Dari et al., 2020). In the last decade, some authors used visible and near infrared RS observations jointly with in situ data collected from inventories to map areas equipped for irrigation (Ambika et al., 2016; Ozdogan & Gutman,

2008). Kumar et al. (2015a) were the first to propose the use of coarse resolution satellite microwave (MW) sensors to detect irrigation. The authors compared different coarse-scale active and passive MW surface soil moisture (SSM) retrievals with SSM



simulations from the Noah LSM (version 3.3; Ek et al., 2003) without activating an irrigation scheme over a continental US domain. Areas where the distributions of model and RS data sets deviated (based on a Kolmogorov-Smirnov test) were assumed to be irrigated. Even though some of the products showed a potential ability to detect irrigation, the authors concluded that the spatial mismatch between the satellite footprint and the irrigated fields, radio-frequency interference (RFI), vegetation, and topography could all deteriorate the accuracy of the results. Similar conclusions were found over the same area by Zaussinger et al. (2019) who compared coarse-scale satellite SSM products with soil moisture predictions from the Modern-Era Retrospective analysis for Research and Applications 2 (MERRA-2) in the absence of precipitation, and Escorihuela and Quintana-Seguí (2016) who additionally compared a downscaled version of the Soil Moisture and Salinity mission (SMOS) SSM to SURFEX LSM simulations. Brocca et al. (2018), Jalilvand et al. (2019) and Dari et al. (2020) used a conceptually different approach with the same coarse scale MW SSM products and estimated irrigation by directly inverting a simple water balance equation (Brocca et al. 2014).

The Copernicus Sentinel-1 satellites (Sentinel-1A and Sentinel-1B) offer a new perspective for agricultural applications, thanks to the finer spatial resolution (up to 10-20 m) of the Synthetic Aperture Radar (SAR) backscatter ($\sigma^0$) data. For instance, Gao et al. (2018) proposed an approach to map irrigated lands over the Urgell region in Catalonia (Spain), and Le Page et al. (2020) proposed a methodology to detect irrigation timing in south-west France comparing the SSM signal at the plot scale, derived using Sentinel-1 $\sigma^0$ and NDVI from Sentinel-2 (El Hajj et al., 2017), with a water budget model forced by Sentinel-2 optical data for the detection of irrigation timing.

Despite the high potential demonstrated by RS in detecting, mapping and quantifying irrigation, the uncertainties of the satellite retrievals, the relatively low revisit time of high resolution active MW products and the too coarse spatial resolution of passive MW products with respect to the mean size of irrigated fields represent main limitations for irrigation information retrieval (Romaguera et al., 2010, La Page et al., 2020). Data assimilation (DA) could reduce some uncertainties by optimally integrating LSM estimates and RS observations. Indeed, the LSM estimates resolve processes at desired spatio-temporal scales, while the RS observations can track in a more realistic way human processes like irrigation and their interactions with the water and energy cycles. Contrasting LSM simulations with RS observations offers an opportunity to correct for unmodeled processes or missed events, such as irrigation (Kumar et al., 2015a; Girotto et al., 2017). More generally, DA of satellite-based observations has shown the potential to update soil moisture (De Lannoy & Reichle, 2016; Kolassa et al., 2017) and vegetation (Albergel et al., 2018; Kumar et al., 2020) and important impacts have been reported over agricultural areas (Kumar et al., 2020).

The assimilation of MW RS observations in LSMs often involves retrieval assimilation. However, assimilating retrievals (i.e., SSM or vegetation optical depth rather than MW brightness temperature (Tb) or $\sigma^0$ measurements) can be problematic as the retrievals may be produced with inconsistent ancillary data (De Lannoy et al. 2016). An alternative solution is to directly assimilate MW observations and equip the LSM with an observation operator that links land surface variables of interest (e.g., soil moisture and vegetation) with RS data. The direct assimilation of MW observations has already been demonstrated successfully for the update of soil moisture by using Tb derived from the SMOS and SMAP missions (De Lannoy et al. 2016, Carrera et al., 2019, Reichle et al. 2019), as well as using radar $\sigma^0$ from ASCAT (Lievens et al., 2017b), and $\sigma^0$ from Sentinel-1



in synergy with SMAP Tb (Lievens et al., 2017a). However, to our knowledge, none of these studies considered the joint updating of soil moisture and vegetation, and none specifically focussed on the performance over irrigated areas. The $\sigma^0$ from Sentinel-1 contains information on both soil moisture (Zribi et al., 2011; Liu and Shi, 2016; Li and Wang, 2018; Bauer-Marschallinger et al., 2018) and vegetation (Vreugdenhil et al., 2018; Vreugdenhil et al., 2020) and assimilating this data could
allow us to update both soil moisture and vegetation in a land data assimilation system and, in doing so, correct for missed irrigation events.

To that end, the LSM needs to be coupled to a backscatter forward model as an observation operator. Different SAR $\sigma^0$ models have been proposed to simulate the backscattering contributions of soil and vegetation (Attema & Ulaby, 1978; Oh, 2004; Zribi et al., 2005; Bai et al., 2015; Baghdadi et al., 2017). Most commonly used, the Water Cloud Model (WCM hereafter) developed
by Attema and Ulaby (1978) is a $\sigma^0$ model that represents the vegetation canopy as a homogeneous cloud containing randomly distributed water droplets. In order to use the WCM as the forward operator in a $\sigma^0$ data assimilation system, it first needs to be calibrated to account for biases between the LSM simulations and the satellite observations. However, calibrating a WCM to simulate $\sigma^0$ over irrigated areas, is not a straightforward process and it represents a key research problem if the same $\sigma^0$ signal is used for the calibration of WCM parameters and later for assimilation and state updating. In fact, if the objective is to assimilate
radar $\sigma^0$ to realistically inform the model about irrigation applications, the WCM parameters have to maintain a certain degree of independence from the irrigation signal contained in the observed $\sigma^0$ as otherwise the assumption of uncorrelated errors between model and observations typical of classical Bayesian-based filters is violated. More specifically, if the LSM provides unrealistic simulations as input (i.e., absence of irrigation), then the WCM calibration with observed $\sigma^0$ would compensate for this bias. This would in turn lead to a biased backscatter model with undesirable calibrated parameters for the subsequent data
assimilation experiments. Therefore, different strategies can be adopted, for instance calibrating the model during non-irrigated periods or over non-irrigated areas, or equipping the LSM with an irrigation module that makes the WCM less constrained by inconsistencies between simulated and observed $\sigma^0$ during irrigation periods. The efficacy of these strategies has so far never been explored.

The main objective of this study is to simulate radar $\sigma^0$ using a LSM coupled with a WCM and to provide solutions and
recommendations for the optimization of the WCM as an observation operator. This is a major stepping-stone towards the development of a reliable system for the assimilation of high-resolution Sentinel-1 $\sigma^0$ observations over irrigated areas. Additionally, we aim at:

1) testing the ability of a sprinkler irrigation system coupled with a LSM to simulate irrigation so as to highlight the potential and limitations of such a tool to optimize a backscatter forward operator over heavily irrigated areas;
2) demonstrating that Sentinel-1 $\sigma^0$ observations contain valuable information to improve both SM and vegetation predictions over irrigated land (i.e., soil moisture and vegetation consistent with human alterations in the water cycle due to intensive irrigation).

The analysis is carried out over the Po river valley, one of the most important agricultural areas in Italy and also one of the more intensively irrigated areas in Europe (water withdrawal in the Po basin is estimated to be 20.5 billion m$^3$/year, of which 16.5





billion of m³/year is withdrawn for irrigation; Po River Watershed Authority, 2006). We use the Noah-MP v.3.6 LSM (Noah-MP hereafter) as part of the NASA Land Information System (LIS) framework together with the WCM from Attema and Ulaby (1978) for the simulation of both $\sigma^0$ vertical send and receive (VV) and vertical send and horizontal receive (VH) polarization. Level-1 Sentinel-1 $\sigma^0$ observations are used to calibrate the WCM at 1-km resolution, using simulated SSM and Leaf Area Index (LAI) estimates from Noah-MP. The WCM is calibrated for a total of four calibration experiments for each polarization: 1) with

or without activating an irrigation scheme within Noah-MP; and, 2) considering two different cost functions. Specifically, we want to demonstrate that activating an -even poor- irrigation scheme is needed to obtain long-term unbiased $\sigma^0$ simulations and uncorrelated errors between the WCM and Sentinel-1 and that the calibration process can be sensitive to different cost functions. The manuscript is organized as follows. Section 2 provides information on the study area, the selected datasets, and methods used for our analysis. Specifically, Sections 2.3 and 2.4 provide a detailed description of the Noah-MP LSM and the WCM.

Section 2.5 describes the cost functions used for the WCM calibration while Section 2.6 is a description of the experimental set-up designed for the calibration. Finally, Section 2.7 provides insights on the Noah-MP and WCM evaluations. Section 3 presents the results, with an assessment of the Noah-MP evaluation, both regional (Section 3.1) and over the test sites (Section 3.2). The WCM calibration and evaluation results are described in Sections 3.3 and 3.4, respectively. We provide discussion in Section 4 while conclusions are reported in Section 5.

## 150   2 DATA AND METHODS

### 2.1 Study area and in situ data

The analysis was carried out over an area of 24,000 km² located within the Po river valley, one of the most important agricultural areas in Europe (Figure 1, left-bottom corner: 44°N, 10.5°W; top-right corner: 45.5°N, 12.2°W). The Po river valley is part of the Po river basin district (~74,000 km²), a mountain-fed catchment which extends from the Alps in the West, to the Adriatic

Sea in the East. The Po district is one of the eight districts mentioned in the Water Framework Directive (WFD, 2000) initiated by the European Commission and has been hit by seasonal drought events which impacted all water use sectors, in particular agriculture (Strosser et al., 2012). The water assessment and impact evaluation of human activities over the Po river valley is thus a topic of major interest, considering the significant requirements from the agricultural management sector.

According to the Köppen-Geiger climate classes (Peel et al., 2007) the study area is classified as "Cfa" (temperate climate,

without dry season and with hot summers). From a geographical point of view, the Po river flows from the west to the east, splitting the area of interest in northern and southern areas, respectively. North of the Po river, the agricultural plain area can be additionally subdivided into the Veneto region to the east and the Lombardy region to the west (Figure 1). Lombardy lands have a high water availability, thanks to the presence of several Alpine lakes and reservoirs (Musolino et al., 2017), as does the Veneto region. Wine cultivation plays an important role, especially in the Garda Lake surroundings (located to the north-west side of

the study area). In the south, the Emilia Romagna region is an agricultural as well as urbanized-industrialized area. Compared to Lombardy and Veneto, Emilia Romagna is much poorer both in water availability and storage capacity, but its irrigation





system is considered the most technologically advanced and efficient in the Po river basin (Musolino et al., 2017). Specifically, it hosts the Canale Emiliano Romagno (CER, https://consorziocer.it/it/, last access 20 May 2021), one of the most important Italian hydraulic systems for agricultural water supply. The main crops in the study region include general summer and winter
crops, orchards (i.e., peach, pear, kiwi), olive groves, and vineyards (https://sites.google.com/drive.arpae.it/servizio-climatico-icolt/home, last access 20 May 2021). The plain area is surrounded by a forested hilly and mountainous area of the Tuscan-Emilian Appennine to the south/south-west.

In situ data were collected over two test sites, located in the Emilia Romagna region:

- The Budrio test site (Figure 1a) is an experimental farm managed by the CER authority and includes two plots of
0.39-0.49 ha. The main crops are maize for field 1 (in yellow) and tomatoes in field 2 (red colour). Daily irrigation data, in mm, were collected for the summer 2015-2016 over field 1, whereas daily irrigation water amounts were collected for the summer 2017 over field 2. Additionally, for field 2, hourly in situ soil moisture data, aggregated here at daily scale, were made available from the Department of Physics and Earth Science of the University of Ferrara. The soil moisture data were derived from an innovative Proximal Gamma-Ray (PGR SM hereafter; Filippucci et al.,
2020, Strati et al., 2018) station, equipped with a 1L NaI(Tl) detector placed at 2.25 m above the ground and a commercial agro-meteorological station (MeteoSense 2.0, Netsen; Strati et al., 2018). The PGR is a nuclear non-invasive and non-contact technique, which allows to overcome the issue connected to in situ point measurements, probing soil moisture with a field scale footprint ($\sim 10^4 \, \text{m}^2$) up to a depth of 30 ~ cm. The quantification of PGR soil moisture is derived from measurements of gamma signals emitted by the decay of $^{40}$K, which is extremely sensitive
to different soil water contents in agricultural soils (for more information on the PGR soil moisture deriving procedure the reader can refer to Baldoncini et al., 2019). Finally, daily rainfall data were collected from the national rainfall network managed by the Department of Civil and Environmental Protection (DPC) of Italy, for the irrigated periods.
- The second test site (Figure 1b) is located around the city of Faenza (hereafter Faenza test site) and has a total extent of 1051 ha, consisting of two different fields. The first one is called San Silvestro (290 ha) and it is located north of
the city. The second one is called Formellino (760 ha), located east to the San Silvestro field and north-east to the city of Faenza. Fruit trees are prevalent on the fields; in particular, pear trees and kiwi dominate the area. The water used for irrigation was provided by CER, at hourly time scale and in mm, for the 2-years time period 2016-2017. Daily rainfall data were collected from the national rainfall network managed from the DPC.

## 2.2 Sentinel-1 $\sigma^0$ and reference remote sensing products

The Copernicus-ESA Sentinel-1 $\sigma^0$ observations were used in this study for the calibration of the WCM. The Sentinel-1 constellation consists of two satellites, Sentinel-1A and Sentinel-1B, launched in 2014 and 2016, respectively. Each satellite carries a Synthetic Aperture Radar (SAR) operating at C-band (5.4 GHz) in the microwave portion of the electromagnetic spectrum. The processing of the ground-range detected (GRD) Interferometric Wide Swath (IW) observations in VV- and VH-polarization was done using Google Earth Engine's Python interface and included standard techniques: precise orbit file


application, border noise removal, thermal noise removal, radiometric calibration, and range-Doppler terrain correction. Furthermore, the $\sigma^0$ observations acquired at $5 \times 20$ m$^2$ resolution were aggregated and projected on the 1 km Equal Area Scalable Earth version 2 (EASE-2) grid (Brodzik et al., 2012). After applying an orbit bias-correction (Lievens et al., 2019), the observations from different orbits, either from Sentinel-1A or -1B and ascending or descending tracks, were combined at the daily time-scale.

Additionally, RS observations were used for the evaluation of the SSM and LAI simulated in Noah-MP LSM for the period 31 March 2015- December 2019:

- The NASA Soil Moisture Active Passive (SMAP; Entekhabi et al., 2010) is an orbiting observatory launched in January 2015 carrying two instruments: a SAR which suffered a failure in early July 2015, and a radiometer measuring Tb at L-band, with a native spatial resolution of 40 km, a revisit time of 2–3 days, and ascending and descending
overpasses at 6:00 PM and 6:00 AM (local time), respectively. For this study, the 9-km SMAP Enhanced Level-2 SSM version 4 (0-5 cm; SMAP L2 hereafter) product was used (O'Neill et al., 2020; Chan et al., 2018). The product is derived from SMAP Level-1B (L1B) interpolated antenna temperatures using the Backus-Gilbert optimal interpolation technique. Both ascending and descending tracks were collected.

- The Metop ASCAT SSM Climate Data Records (CDR) H115 and its extension H116 are provided by the European
Organization for the Exploitation of Meteorological Satellites (EUMETSAT) Support to Operational Hydrology and Water Management (H SAF; http://hsaf.meteoam.it/, last access 20 May 2021). The SSM is retrieved from $\sigma^0$ using a change detection algorithm (Wagner et al., 2013), and is characterized by a spatial sampling of 12.5 km and a temporal resolution of one to two observations per day, depending on the latitude.

- The PROBA-V LAI is derived from the PROBA-V satellite mission (Francois et al., 2014; Dierckx et al., 2014) and
provided by the Copernicus Global Land Service programme (CGLS, https://land.copernicus.eu/global/). The CGLS product at 1 km spatial resolution and 10-day (dekadal) temporal resolution is developed based on the work by Verger et al. (2014).

In order to compare Noah-MP simulations and reference data at the same spatial resolution, Sentinel-1 observations ($\sigma^0$-VV and -VH), as well as ASCAT SSM, SMAP L2 SSM and PROBA-V LAI were extracted over the study domain (left-bottom corner:
44°N, 10.5°W; top-right corner: 45.5°N, 12.2°W) and re-gridded over the LIS grid domain (0.01°) using the nearest-neighbour approach.

### 2.3 Land surface and irrigation modelling

### 2.3.1 Noah-MP v.3.6

The analysis was carried out using the Noah-MP (Niu et al., 2011) LSM, running within NASA's LIS 7.2 version (Kumar et al.,
2008). LIS is a software framework for terrestrial hydrology modelling and DA, which supports different LSMs that can be conditioned on multiple remote sensing products from active and/or passive microwave sensors. The Noah-MP LSM, which



was chosen for this study, is an evolution of the baseline Noah LSM (Mahrt and Ek, 1984; Chen et al., 1996; Chen and Dudhia, 2001) wherein main improvements and augmentations are: 1) the presence of four soil layers; 2) up to three snow layers; 3) one canopy layer which allows to dynamically simulate the vegetation and to compute separately the ground surface temperature; 4)

a two-stream radiation transfer scheme based on the canopy layer sub-grid scheme; 5) a Ball-Berry type stomatal resistance scheme; 6) and finally, a simple groundwater model with a TOPMODEL-based runoff scheme (Niu et al., 2005, 2007). The model was set up selecting four soil layers at depths 0–10, 10–40, 40–100 and 100–200 cm, a dynamic vegetation model with a Ball-Berry type canopy stomatal resistance model (Ball et al., 1987), and TOPMODEL-based runoff.

The parameterization followed the recommended options provided in the LIS documentation
(https://modelingguru.nasa.gov/docs/DOC-2634). A model time step of 15 minutes and a 6 hours output interval were selected together with a spatial resolution of 0.01°. The meteorological forcings used for running Noah-MP LSM were obtained from MERRA-2 (Gelaro et al. 2017). The MERRA-2 original spatial resolution of 0.5°x0.625° was re-mapped to 0.01° through bilinear interpolation. Land model data and parameters were pre-processed and adapted to the LIS longitude/latitude projection using the Land Surface Data Toolkit (LDT; Arsenault et al., 2018) in order to run Noah-MP at the chosen spatial resolution.

For this study, the default LIS Land Cover (LC) map from the University of Maryland (UMD) global land cover product (Hansen et al., 2000) based on the Advanced Very High Resolution Radiometer (AVHRR) data was replaced with the 2015 global LC map, available from the CGLS at 100 m spatial resolution (Buchhorn et al., 2020; available at https://land.copernicus.eu/global/products/lc , last access 20 May 2021). The CGLS provides Dynamic Land Cover Layers at 100 m spatial resolution (CGLS-LC100), obtained by combining information derived from the vegetation instrument on board

the PROBA-V satellite, a database of high-quality LC reference sites, and several ancillary datasets. For a more detailed explanation of the LC maps generation process we refer to the Algorithm Theoretical Basis Document (ATBD; Buchorn et al., 2020). The 23 classes of the PROBA-V LC map were reclassified to the 14 classes used in the UMD-AVHRR classification supported by LIS. Additionally, the LC map was regridded at 0.01° (Figure 2a) by identifying the most representative class over each LIS grid cell. For additional information on the reclassification process, we refer the reader to Table S1 in the

Supplementary Material section. Similarly, the default FAO Soil Map (FAO Soil Map of the World, 1971) was replaced by the Harmonized Soil World Database (HWSD v1.21, 1 km; Figure 2b) and mapped to 5 soil classes over the study region. Other model pre-processed parameters inputs were: i) the Shuttle Radar Topography Mission elevation data (SRTM30, 30 m spatial resolution); 2) climatological global Greenness Vegetation Fraction (GVF) data (0.144°; Gutman and Ignatov, 1998) derived from 5 years (1985-1989) of normalized difference vegetation index (NDVI) data from the AVHRR (Miller et al., 2006); 3) a

snow-free albedo and a Noah-specific maximum snow albedo product from NCEP (original resolution 1° and regridded); and finally, 4) soil, vegetation, and other general parameter tables for Noah-MP from the LIS official Data Portal (https://portal.nccs.nasa.gov/lisdata_pub/data/, last access 20 May 2021).





### 2.3.2 Irrigation modelling

The ability of Noah-MP to dynamically simulate the vegetation and the option to activate irrigation are particularly important
considering an extensively irrigated area such as the Po river valley. Indeed, in a recent study by Nie et al. (2018), Noah-MP
was coupled with a sprinkler irrigation scheme (Ozdogan et al., 2010b) (where irrigation is applied as supplementary rainfall),
which requires three pieces of information:

- the irrigation location, only occurring over potentially irrigated croplands (expanding over grassland if the intensity exceeds the gridcell's total crop fraction). This information is extracted from a LC map associated with an additional
dataset providing information on the percent of irrigated area per grid cell. In this study, the reclassified PROBA-V LC map was coupled with the information contained in the 500 m Global Rain-fed, Irrigated and Paddy Croplands data set (GRIPC; Salmon et al., 2015);

- the timing of irrigation, which is determined by checking the start and end of the growing season based on a GVF threshold, separately at each grid cell. Following Ozdogan et al.(2010b), we set this threshold to 40% of the GVF;

- the amount of water used for irrigation. This quantity is derived from the root zone soil moisture (RZSM) availability (MA) as MA=(RZSM-SM$_{WP}$)/(SM$_{FC}$-SM$_{WP}$) where RZSM is the current RZSM, SM$_{WP}$ is the wilting point, and SM$_{FC}$ is the field capacity. When the MA falls below a user-defined threshold, irrigation is triggered and the quantity is defined by calculating the amount of irrigation needed to raise the RZSM to the SM$_{FC}$. For this study, the MA threshold was defined as the 50% of SM$_{FC}$ as in Ozdogan et al. (2010b). MA is calculated at each time step but the irrigation is only
applied between 06:00 and 10:00 LT. Following Ozdogan et al. (2010b), this time frame is typically chosen by farmers to reduce evaporative losses. In this context, the maximum rooting depth becomes a crucial information to compute the amount of irrigation water. This information is related to an assigned crop type, cultivated over the study area, through a maximum rooting depth table. Considering the high crop variability over the Po river valley as well as the lack of high resolution dynamic crop maps for the entire study area, a generic crop type with 1 m root depth was selected for the
irrigation simulations. The reference rooting depth was verified to be feasible over the study area based on the European Soil Data Centre (ESDAC, available at https://esdac.jrc.ec.europa.eu/content/european-soil-database-derived-data, last access 20 May 2021) rooting depths map (Figure S1 in the Supplementary Material).

### 2.4 Water Cloud Model

The WCM allows to simulate the top-of-vegetation σ$^0$ as a function of SSM and vegetation, using empirical fitting parameters.
σ$^0$ is modeled as the sum of the backscatter from the vegetation ($\sigma^0_{veg}$, in dB) and from the bare soil ($\sigma^0_{soil}$, in dB), attenuated
by the t$^2$ coefficient that describes the two-way attenuation from the vegetation layer. Scattering interactions between the
ground and the vegetation are not accounted for. As reported in Baghdadi et al. (2018), for a given polarization pq (i.e., VV
and VH), the WCM can be written as follows:





$$\sigma^0_{pq} = \sigma^0_{veg,pq} + t^2_{pq}\sigma^0_{soil,pq} \tag{1}$$

where:

$$\sigma^0_{veg,pq} = A_{pq}V_1\cos\theta(1 - t^2_{pq}) \tag{2}$$

$$t^2_{pq} = \exp(\frac{-2B_{pq}V_2}{\cos\theta}) \tag{3}$$

$$\sigma^0_{soil,pq} = C_{pq} + D_{pq} \cdot SSM \tag{4}$$

Equations 2 and 3 describe the vegetation-related terms. $V_1$ and $V_2$ represent two bulk vegetation descriptors, the first one accounting for the direct vegetation $\sigma^0$, and the second one representing the attenuation. $A_{pq}[-]$ and $B_{pq}[-]$ are the two related fitting parameters. Common vegetation descriptors used in previous studies are the Vegetation Water Content (VWC, Paloscia et al., 2013), the NDVI (El Hajj et al., 2016; Li and Wang, 2018) and LAI (Kumar et al., 2015b; Bai and He, 2015), while $\theta$ represents the incidence angle, which is assumed to be 37° for Sentinel-1. Many studies assumed $V_1=V_2$ represented by the

unique vegetation descriptor (see Lievens et al, 2017b, Baghdadi et al. 2017, Li and Wang, 2018).

Equation 4 describes the soil-related term. Following the work by Lievens et al. (2017b), the $\sigma^0_{soil}$ can be described, in a simple linear approach, as a function of the SSM. There are several semi-empirical models (e.g., the Oh model; Oh et al., 1992) or theoretical models (e.g., the Integral Equation Model (IEM), Fung, 1994) which describe the scattering processes related to the bare soil, but their application as a forward operator coupled to an LSM has two main limitations: the first one lies in the

difficulty in retrieving soil roughness values over extended reference areas required to parameterize these models; the second one is their saturation of $\sigma^0$ in moist conditions which causes low variability in simulated $\sigma^0$ if the LSM soil moisture simulations are biased wet (for more information see Lievens et al., 2017b). Those limitations justify the use of a linear fitted approach. In Equation 4, the C and D parameters (here fitted in dB and dB/m³/m³, respectively, but $\sigma^0_{soil}$ is transformed back to the linear scale in Equation 1) describe the linear relation between $\sigma^0_{soil,pq}$ and SSM. Those parameters, as well as A and

B (-), need to be calibrated separately for each polarization.

**2.5 Calibration algorithms**

We considered two different objective functions to optimize the A, B, C and D parameters:

- a Bayesian solution, which minimizes the Sum of Squared Errors (SSE) between $\sigma^0$ observations from Sentinel-1 and WCM simulations. The SSE Bayesian calibration solution aims at identifying the optimal parameter vector $\boldsymbol{\alpha}$ which
maximizes the probability of the resulting $\sigma^0$ simulations $p(\hat{y}^-) = p(\hat{y}^-|\alpha)p(\alpha)$, where $p(\alpha)$ is the prior parameter distribution and $p(\hat{y}^-|\alpha)$ is the likelihood. Starting from the assumption of an independent and identically distributed





normal error model, the posterior probability can can be maximized by maximizing:

$$p(\hat{y}^-|\alpha)p(\alpha) = \prod_i^{N_i}\left\{\frac{1}{s_i\sqrt{2\pi}}exp\left(-\frac{(\hat{y}-\hat{y}^-)_i^2}{2s_i^2}\right)\right\} \cdot \prod_j^{N_\alpha}\left\{\frac{1}{s_j\sqrt{2\pi}}exp\left(-\frac{(\alpha_0-\alpha)_j^2}{2s_j^2}\right)\right\} \qquad (5)$$

i.e., the combination of the likelihood and a prior parameter constraint. The latter helps in reducing problems of equifinality. In Equation (5), $\hat{y}$ represents the observed $\sigma^0$, $\hat{y}^-$is the simulated $\sigma^0$, $i$ is the timestep and $s_i$ is the standard deviation of the residual differences between the observed and simulated $\sigma^0$ values for $N_i$ time steps. $N_\alpha$ is the number of parameters to be calibrated, $\alpha_0$ is the prior parameter constraint and the parameter deviation is limited by $s_j^2$, the variance of a uniform distribution $s_j^2 = \left(\alpha_{max,j} - \alpha_{min,j}\right)^2/12$ with determined boundaries of the parameters $[\alpha_{min},\alpha_{max}]$. The maximum likelihood solution is found by minimizing the following cost function $J$:

$$J = \sum_i^{N_i}\left\{ln(s_i) + \frac{(\hat{y}-\hat{y}^-)_i^2}{2s_i^2}\right\} + \sum_j^{N_\alpha}\left\{\frac{((\alpha_0-\alpha)_j^2}{2s_j^2}\right\} = J_0 + J_\alpha \qquad (6)$$

where $s_i$ is assumed to be constant in time and represented by a target accuracy of 1 dB, leaving the SSE in the first term of $J_0$ to minimize. The second term ($J_\alpha$) constrains the optimal solution by avoiding strong deviations from initial parameter guesses.

- a solution that maximizes the Kling-Gupta Efficiency (KGE; Gupta et al., 2009). Even though this objective function does not ensure Bayesian optimality, it is a widely used metric which could help to better tune the dynamic $\sigma^0$ behaviour:

$$KGE = 1 - \sqrt{(r-1)^2 + \left(\frac{<\hat{y}^->}{<\hat{y}>} - 1\right)^2 + \left(\frac{s[\hat{y}^-]/<\hat{y}^->}{s[\hat{y}]/<\hat{y}>} - 1\right)^2} \qquad (10)$$

The *KGE* formulation embeds three terms: 1) the first term accounting for the Pearson Correlation (Pearson-R) between the observed ($\hat{y}$) and simulated ($\hat{y}^-$) $\sigma^0$ time series; 2) a second term accounting for the bias, where the long-term mean is represented as <.>; and finally, 3) a term accounting for the variability of the simulated and observed signal through the use of the standard deviation s[.]. *KGE* = 1 indicates a perfect agreement between simulations and observations. Note that *KGE* redistributes the weight of the bias, variance and correlation components, compared to *J* in Equation 6, which can help in reducing differences between simulated and observed $\sigma^0$ also in terms of temporal dynamics during the calibration. On the other hand, in the *KGE* cost function parameters are not constrained by prior values $\alpha_0$. This could possibly result in overfitting and a larger prediction uncertainty.

The Particle Swarm Optimization (PSO; Kennedy and Eberhart, 1995) was used to minimize *J* and maximize *KGE*. For our case study the PSO parameters were set as in De Lannoy et al. (2013).





## 2.6 Experimental setup

Building an optimal DA system able to correct for the poor parameterization of irrigation within LSMs via the ingestion of
radar $\sigma^0$ requires the minimization of the impact of the irrigation signal already contained in $\sigma^0$ observations on the WCM parameters, while simultaneously guaranteeing long-term unbiased $\sigma^0$ simulations compared to observations. Here we tested the hypothesis that this can be only achieved by activating irrigation in the LSM.

To that end, we considered two different experiment lines (referred to as *Natural* and *Irrigation,* respectively) that produced a total of eight different $\sigma^0$ simulation runs (see Figure 3). The *Natural* experiment line differs from the *Irrigation* line by the
activation of an irrigation module in Noah-MP, and both are subjected to the calibration algorithms described in Section 2.5. The *Natural* line was used as a diagnostic experiment against which to compare *Irrigation,* which, according to our initial hypothesis, should minimize the impact of the irrigation signal contained in the $\sigma^0$ observations on WCM parameters.

As a first step, a model spin up was performed, starting in January 1982 and ending in December 2014. Then, a study period from January 2015 to December 2019 was selected for the different model runs based on the availability of the processed
Sentinel-1 $\sigma^0$ and reference irrigation data (see Sections 2.1 and 2.2). Daily surface model and irrigation outputs were produced. Considering that the main source of irrigation in the Po river valley is related to surface water abstraction, the sprinkler irrigation scheme did not account for groundwater withdrawals (see Nie et al., 2018).

The A, B, C, and D parameters of the WCM (see section 2.4) were fitted for each grid cell based on Sentinel-1 $\sigma^0$ VV and VH observations separately, during the period January 2017 - December 2019. Both the calibration using the SSE with prior
constraint (Bayesian *J*) and the KGE were applied to the *Natural* and *Irrigation* runs providing eight different experiments named *J-VV Natural*, *J-VH Natural*, *J-VV Irrigation*, *J-VH Irrigation*, *KGE-VV Natural*, *KGE-VH Natural*, *KGE-VV Irrigation* and *KGE-VH Irrigation.*

Lower and upper boundaries as well as prior guess values of the WCM parameters were defined based on the work of Lievens et al. (2017b) and on a sensitivity analysis (not shown here). The selected values are displayed in Table 1. Finally, it should be
noted that all the calibration experiments were realized by considering daily values of $\sigma^0$ simulations and observations.

**Table 1: Lower boundaries (LB), upper boundaries (UB), and prior guess values of the WCM parameters for both VV and VH polarization**

|  | A-VV[-] | A-VH[-] | B-VV[-] | B-VH[-] | C-VV[dB] | C-VH[dB] | D-VV[dB/m³/m³] | D-VH[dB/m³/m³] |
|---|---|---|---|---|---|---|---|---|
| UB | 0.4 | 0.4 | 0.4 | 0.4 | -10 | -10 | 80 | 80 |
| LB | 0 | 0 | 0 | 0 | -35 | -35 | 15 | 15 |
| GUESS | 0 | 0 | 0 | 0 | -20 | -30 | 40 | 40 |



### 2.7 Noah-MP LSM and WCM evaluations

The validation aims at i) evaluating the performance of Noah-MP in simulating irrigation, soil moisture, and vegetation and the ability of the WCM to simulate radar $\sigma^0$, and ii) unveiling the information about irrigation contained in Sentinel-1 radar $\sigma^0$ in order to assess its potential to improve both soil moisture and vegetation representation within Noah-MP.

The evaluation was carried out both on the regional scale (i.e., over the entire study area) and at two selected sites, Budrio and Faenza, where irrigation data were available. We compared Noah-MP (with and without using the irrigation module) SSM and

LAI simulations with satellite SSM from ASCAT and SMAP, and LAI from PROBA-V, respectively, during the period 2015-2019. Furthermore, these land surface simulations were compared to Sentinel-1 $\sigma^0$ to understand how much of the SSM and LAI signal was captured by Sentinel-1.

As the irrigation timing is often driven by the stakeholders' turns to withdraw water and by water availability rather than by the conditions of the soil and crops themselves, the comparisons between simulated SSM and satellite SSM were carried out

by aggregating the two variables over a bi-weekly time window. On the other hand, the LAI from Noah-MP was aggregated to ten-daily values in order to match the dekadal PROBA-V LAI values. We used the Pearson-R for SSM and LAI evaluation. For LAI, we also considered the ratio bias, i.e., the ratio between the long-term mean of the simulations and the long-term mean of observations. This additional score was used to provide a further evaluation of the ability of the Noah-MP to simulate crop phenology during the irrigated vs non-irrigated periods so as to not rely solely on the evaluation of temporal dynamics,

which, due to the uncertainty in the Noah-MP crop type parameterization, could be affected by time shifts in the LAI climatology. This parameterization uncertainty comes from the lack of knowledge of the spatial crop type information and is difficult to be reduced without additional information. Our assumption is that radar $\sigma^0$ assimilation can also correct for this with future data assimilation.

Following Vreugdenhil et al. (2018) and Vreugdenhil et al. (2020), Noah-MP LAI and PROBA-V LAI were also compared

with the Sentinel-1 $\sigma^0$ VH/ $\sigma^0$ VV cross ratio (CR), which was demonstrated to have a high agreement with the vegetation signal. Though the $\sigma^0$ VH was demonstrated to increase with the vegetation signal (Macelloni et al., 2001), the CR will be more sensitive to vegetation changes as the ratio is less sensitive to changes in soil moisture and soil-vegetation interaction (Veloso et al., 2017; Vreugdenhil et al., 2020).

To evaluate WCM simulations, we used bi-weekly values of $\sigma^0$ simulations and observations considering a two-years period

independent from the calibration period: 2015-2016. Statistical metrics such as grid-based temporal Pearson-R, KGE, and bias were calculated between Sentinel-1 $\sigma^0$ and calibrated WCM simulations. The analysis of the parameters was restricted to the cropland area as no difference between our experiment lines exists over other land cover types (i.e., the irrigation module is active only over grid points classified as crop).



## 3 RESULTS

### 3.1 Noah MP regional evaluation

Figure 4 shows maps of the Pearson-R between bi-weekly Noah-MP SSM *Natural* and *Irrigation* simulations and bi-weekly ASCAT and SMAP L2 SSM retrievals, respectively, for April 2015 to December 2019. The Noah-MP SSM *Irrigation* run provides a higher agreement with both satellite SSM data sets compared to the *Natural* run. Indeed, the median Pearson-R between SMAP L2 SSM and Noah-MP SSM increases from 0.68 to 0.73, for the *Natural* run (Figure 4a) and the *Irrigation* run (Figure 4b), respectively. A similar improvement can be observed considering the ASCAT reference SSM, with an improvement in the median Pearson-R of 0.08 when irrigation is activated in the model (from 0.7 to 0.78; Figure 4e). Areas characterized by higher correlation when irrigation is simulated are represented in blue in the Pearson correlation difference map of Figure 4f (obtained by subtracting the map in Figure 4d from the map in Figure 4e). Almost all cropland areas are characterized by a higher agreement between observations and simulations for the *Irrigation run*. Note that for the evaluation of Noah-MP against SMAP, we relaxed retrieval quality flags, which would otherwise mask out almost the entire study area. The Supplementary material (Figure S2) shows the coverage when using the recommended quality flags.

The evaluation of the LAI simulation was limited to the regional scale analysis due to a lack of in situ vegetation data over the selected test sites. The comparison between dekadal values of Noah-MP LAI, from both model runs, and the PROBA-V LAI product was carried out over the reference period January 2015 to October 2019 using the temporal Pearson-R and the ratio bias, shown in Figure 5.

Figure 5a and 5b show that the Pearson-R for vegetation has a lower median value of 0.67 when irrigation is simulated in Noah-MP, whereas this value equals 0.72 for the *Natural* run. The difference between the two Pearson-R maps is shown in Figure 5c, providing evidence of the areas facing a deterioration of the performance in terms of Pearson-R related to the *Irrigation* run. This deterioration is particularly strong over cropland areas south to the Po river (red colour) while the northern area also shows grid cells where the performance improves (blue colour).

By contrast, the ratio bias evaluation score (Figures 5d, 5e, 5f) highlights an improvement in long-term mean vegetation simulations when irrigation is included (Figure 5e). Here the optimal condition is represented by a ratio bias equal to 1 when the mean of the simulated LAI is equal to the mean of the observed LAI. In this context, Figure 5d displays ratio bias values lower than one over a large central triangle-shaped cropland area and median ratio bias value of 0.73, highlighting an underestimation of the LAI simulation related to the *Natural* run. Conversely, Figure 5e shows ratio bias values close to one when irrigation is simulated over an extended cropland area and a median bias value of 0.99. The improvement given by the *Irrigation* run is emphasized in Figure 5f where the histograms of the ratio bias distributions related to both model runs show the higher performance of the *Irrigation* run (red) compared to the *Natural* run (blue) for which the distribution is more skewed to the zero value.



## 3.2 Noah MP site evaluation

The Noah-MP SSM was evaluated at the Budrio test site field 2 (Figure 1a), using the daily reference PGR SM for the year 2017. Comparisons between the SSM simulations of the *Natural* and *Irrigation* runs with in situ PGR SM are shown in Figure 6a, while daily observed irrigation and rainfall data are compared with daily irrigation simulations in Figure 6b. Soil moisture data are plotted at their original temporal resolution (i.e., daily) to illustrate an issue related to the irrigation timing: SSM simulations in Figure 6a show the ability of the sprinkler irrigation scheme to simulate irrigation in the summer season, but there is an inevitable problem in reproducing the correct timing and magnitude of irrigation. Indeed, the total amount of simulated irrigation is 604 mm for the 2017 summer season, which overestimates the total amount of observed irrigation, being 349.5 mm. Furthermore, the model simulations not only miss irrigation, but also suffer from erroneous precipitation input, such as on the 11th of July 2017, where the observed precipitation event in the growing season is not found in the model SSM simulations. In any case, bi-weekly Pearson-R between simulated SSM and in situ PGR SM are higher for the *Irrigation* run than for the *Natural* run (0.54 vs 0.42) suggesting the benefit in activating irrigation.

For the Budrio field 1 test site (Figure 1a), two summer seasons of irrigation data were available. To assess the irrigation information contained in Sentinel-1 $\sigma^0$ observations (and the potential added value for a forthcoming DA experiment) we compared bi-weekly values of Sentinel-1 $\sigma^0$ VV and VH with SSM estimates from both the *Natural* run and *Irrigation* run (Figure 7a) for this site. Information related to the irrigation periods are shown in Figure 7c, where irrigation observations and irrigation simulations from Noah-MP are compared. Figure 7a indicates that the SSM simulations are better reflected in the Sentinel-1 $\sigma^0$ VV than $\sigma^0$ VH data, particularly when irrigation is simulated (orange line). The SSM estimates from the *Natural* run (light blue line) agree poorly with the Sentinel-1 data, with Pearson-R values equal to 0.14 and -0.13 for the $\sigma^0$ VV (blue dots) and $\sigma^0$ VH (cyan dots), respectively. When irrigation is simulated, the $\sigma^0$ VV data better follow the modelled SSM signal (Pearson-R of 0.36) especially during the summer irrigation season when the backscatter signal remains higher and stable. On the other hand, $\sigma^0$ VH seems to provide poor performances also when irrigation is simulated with a Pearson-R value equal to -0.03.

In Figure 7b, the Sentinel-1 $\sigma^0$ CR (VH/VV) is compared with Noah-MP LAI from the *Natural* run (light-blue line) and *Irrigation* run (orange line). The performance in terms of Pearson-R decreases from 0.8 to 0.51, when the irrigation is simulated. This is due to a time shift of the Noah-MP LAI growing season in the *Irrigation* run. PROBA-V LAI (in green) was additionally compared with the Sentinel-1 CR (blue dots) showing a Pearson-R of 0.88. The higher agreement between the RS products (Sentinel-1 and PROBA-V) highlights the strong relation between the $\sigma^0$ CR and the vegetation signal, suggesting a potential benefit of Sentinel-1 assimilation to correct the simulated vegetation phenology.

Finally, Figure 7c shows a comparison between 15-days accumulated mm of simulated irrigation (in orange) and observed irrigation (in green). The Pearson-R is equal to 0.77, indicating that the sprinkler irrigation scheme can provide acceptable irrigation estimates at this temporal resolution though absolute irrigation amounts are overestimated.



### 3.3 WCM calibration

The WCM parameters A and B (vegetation parameters), and C and D (soil parameters) were calibrated for each grid cell separately during the reference period January 2017 to December 2019 (Figure 3), using daily $\sigma^0$ simulations and observations.

The calibrated parameters related to the entire study area for each of the eight experiments are shown in Figure 8 where the blue left parts of the violin plots identify experiments of the *Natural* run, while the orange right parts of the violin plots are related to the *Irrigation* run.

Generally, the *J*-calibration provides parameter distributions closer around their prior guess as compared to the *KGE*-calibration for which the distributions are often multimodal, especially for the C and D parameters (i.e., Figure 8d, 8h). This

is due to the prior parameter penalty, which is included in the Bayesian solution but not in the *KGE*. In general, the calibration of the two functions using the *Natural* run provides wider distributions between lower and upper boundaries for the A vegetation parameter with a high number of grid cells characterized by A-values higher than 0.1 (see *KGE-VV Natural* and *J-VV Natural* experiments in Figures 8a and 8e respectively). Conversely, the *Irrigation* run provides A distributions more skewed to the lower boundary (being also the guess value in each calibration experiment), with a smaller number of grid cells

characterized by high A values compared to the *Natural* run. In a preliminary sensitivity study (not shown), we observed that high values of the vegetation parameters A and B, as obtained for the *Natural* run, have the tendency to generate high peaks in the simulated $\sigma^0$ during the growing season. Indeed, in the summer, the SSM *Natural* signal is low and not consistent with the Sentinel-1 $\sigma^0$, which observes irrigation. In order to follow the temporal dynamics of the Sentinel-1 $\sigma^0$, the calibration algorithms attribute a relatively higher weight (higher A values) to the LAI than to SSM to compensate for the underestimated

SSM in the *Natural* run. By contrast, the *Irrigation* run provides vegetation parameter distributions more skewed to the lower boundaries (see also Section 3.4.2). Also the C and D parameter distributions show more realistic values using *Irrigation* run input data, and feature a better sensitivity of $\sigma^0$ to soil moisture dynamics. This is true especially when using the *J* cost function (see parameters distributions for the *J-VV Natural* and for the *J-VV Irrigation* experiments in Figures 8g and 8h), which results in more uniform calibrated C and D distributions for the *Irrigation* simulations (esp. in VV polarization), whereas the mode

of the C and D parameter distributions for the *Natural* experiments is more shifted to the upper and lower boundaries, respectively.

The different polarization experiments generally provided similar distributions for the vegetation A and B parameters and the D soil parameter. The largest differences between the VV and VH polarizations are identified for the C parameter distributions. This is due to the lower $\sigma^0$ signal associated with the VH polarization. Indeed, Figure 8c and 8g are characterized by higher

values of the C in VV polarization, as compared to the distributions for VH polarization in Figures 8k and 8o. In the latter, the C-VH distributions are generally more skewed to the lower boundary of the parameters, with median values closer to the defined guess parameter value.

100


### 3.4 WCM evaluation

#### 3.4.1 Regional evaluation

The regional evaluation of the calibration experiments was carried out during the period January 2015 to December 2016 for agricultural areas within the study domain (almost 15,000 km$^2$), by comparing bi-weekly $\sigma^0$ simulations with Sentinel-1 $\sigma^0$ in terms of Pearson-R, KGE, and bias. The distribution of the evaluation metrics for the eight experiments is shown in Figure 9. A comparison of the metrics for the *Irrigation* and *Natural* runs confirms better results when irrigation is activated, with violin plots skewed towards more positive values for both KGE and Pearson-R. When stratified by the cost function, the Pearson-R

distribution in Figure 9a-d indicates slightly higher performance for the *KGE* (Figures 9a and 9c) than for *J* (Figure 9b and 9d). In terms of the KGE score, simulations are naturally closer to the observations when the *KGE* cost function is used. On the other hand, in terms of bias, generally better performances are found when the Bayesian solution is used (Figures 9i-l). The latter is particularly evident for the VH polarization when comparing the *KGE-VH* and *J-VH* experiments (Figure 9k and 9l). The VH simulations exhibit a better performance in the *Irrigation* run than VV simulations (Figure 9c-d and Figure 9a-b).

Indeed, considering all the statistical scores, the VV polarization is characterized by more similar distributions between the *Natural* and *Irrigation* run for both cost functions. This suggests a higher sensitivity of the VH polarization to the change of vegetation introduced by irrigation, confirming the Sentinel-1 $\sigma^0$ VH to be strongly influenced by irrigation as witnessed by the larger score improvement obtained for the calibration experiments *KGE-VH Irrigation* (Figure 9g) and *J-VH Irrigation* (Figure 9h), compared to the *Natural* runs experiments.

In summary, i) VH polarization is more sensitive to the change in the cost function and input data (*Irrigation* or *Natural* run) than VV polarization likely due to its higher sensitivity to vegetation change (Vreugdenhil et al., 2018; Macelloni et al. 2001) which, in the area, is related to the crop development after irrigation, ii) the combination of *J* with activation of the irrigation scheme is able to provide the best unbiased estimates of simulated $\sigma^0$ for both VV and VH (*J-VV Irrigation* and *J-VH irrigation* experiments) at the price of generally lower correlations (compared to the *KGE* cost function). This is, however, beneficial for

DA as it minimizes the chance of potential error cross correlation between model estimates and observations. Indeed, the match of the temporal dynamic of the signals induced by the correlation term is stronger in the *KGE* than in *J*, which additionally includes a parameter constraint. The higher weight of the correlation in the *KGE* cost function can negatively impact the parameter calibration even when irrigation is turned on in Noah-MP because the simulated irrigation applications are in general not temporally consistent with those seen by Sentinel-1 (see Figure 6).

#### 3.4.2 In situ evaluation

The WCM simulations are further analysed in detail at the Faenza test site (specifically for the San Silvestro field), because it has a larger extent than the Budrio site (see Figure 1), although the same overall conclusions were found for Budrio. Figure 10 shows simulated and observed $\sigma^0$ time series for the different experiments highlighted in Figure 3, and Table 3 summarizes the statistics (i.e., Pearson-R, KGE and bias) of each experiment.



The agreement between simulated and observed $\sigma^0$ measured by the Pearson-R and KGE in Table 3 generally gives better performances after calibration with the *KGE* cost function than with the *J* cost function, except for the *KGE-VH Irrigation* experiment in terms of Pearson-R (Figure 10b). For the latter, we can observe a Pearson-R of 0.33 against 0.37 for *J-VH Irrigation* (Figure 10d). On the other hand, in terms of bias the cost function *J* significantly outperforms the calibration with *KGE* in all experiments with surprisingly comparable values between *Natural* and *Irrigation* runs (Table 2).

One undesirable feature of *Natural* runs is the presence of high $\sigma^0$ peaks during the summer, clearly detectable over the Faenza test site in both VV and VH polarization which are less evident in the *Irrigation* run (see Figure 10b and 10d). A similar behaviour was found for Budrio (not shown). These peaks are likely attributed to the poor estimation of model vegetation parameter values, previously discussed in section 3.3, when the WCM attempts to compensate for bias in SSM and vegetation input, i.e., input that is not consistent with observations over irrigated areas. This is particularly true for the *KGE* calibration, 

which does not use a prior parameter constraint. In contrast, the *J* calibration still provides reasonable $\sigma^0$ simulations that are closer to the ones of the *Irrigation* run due to the Bayesian technique itself.

**Table 2: Results of the site WCM evaluation considering the test site Faenza San Silvestro for each WCM experiment**

|  | *KGE-VV Natural* | *KGE-VV Irrigation* | *J-VV Natural* | *J-VV Irrigation* | *KGE-VH Natural* | *KGE-VH Irrigation* | *J-VH Natural* | *J-VH Irrigation* |
|---|---|---|---|---|---|---|---|---|
| *Pearson-R [-]* | 0.14 | 0.32 | 0.02 | 0.3 | 0.39 | 0.33 | 0.28 | 0.37 |
| *KGE [-]* | 0.05 | 0.31 | 0.006 | 0.28 | 0.13 | 0.33 | 0.28 | 0.16 |
| *Bias [dB]* | -0.54 | 0.81 | 0.26 | 0.18 | -0.7 | -0.71 | 0.02 | 0.07 |

## 4 DISCUSSION

### 4.1 Noah-MP irrigation modelling

The Noah-MP LSM, used as input for the WCM calibration, was evaluated in two configurations, either with a sprinkler irrigation scheme activated or without irrigation (i.e., *Irrigation* run and *Natural* run). Although not all of the Po river valley is irrigated by sprinkler systems, it most likely still leads to more realistic LSM simulations than not considering irrigation at all.

The main limitation found in the irrigation simulations was related to the irrigation timing and magnitude that was inconsistent with observations. Although this finding is based on only a single study site, it is very likely that it is a widespread issue within the study area for several reasons. In LSMs, the irrigation application is driven by the RZSM availability and consequently by the soil type and the rooting depth parametrizations. Moreover, it is also influenced by the accuracy of the meteorological



forcings (especially precipitation, Reichle et al. 2017) which can determine errors in the soil moisture representation. The main
reason, however, is likely that irrigation is often the result of subjective farmer decisions rather than objective rules based on
the soil state and crop conditions. In theory, the irrigation timing issue could be partly solved by using temporally consistent
high-resolution crop maps which should provide a more realistic information of crop phenology and rooting depth. However,
in practice, this is unfeasible over many areas of the world given the absence of this information on a large scale. Also, given
that irrigation applications are mainly linked to unmodelled processes like rotation schedules for farmers to withdraw water,
the correct simulation of the timing can be unsolvable when using models only.

Despite the potential problems related to the unrealistic assumptions in the simulation of irrigation, our results demonstrated
that even the use of simple irrigation schemes within Noah-MP can be beneficial. In the regional evaluation, SSM simulations
of the *Natural* and *Irrigation* runs were compared with RS SSM from SMAP and ASCAT (Figure 4) on a bi-weekly temporal
scale. For both products, we found large improvements in temporal Pearson-R when irrigation was simulated, suggesting that
the activation of irrigation modelling provides more realistic SSM estimates. Our findings further confirm the potential of
coarse resolution datasets for providing irrigation-related information over intensively irrigated and relatively large agricultural
areas, as was shown by Kumar et al. (2015a).

While the impact of irrigation was clear in terms of SSM, the regional evaluation of the simulated LAI against the PROBA-
V-based LAI provided contradicting results. In this case, the Pearson-R analysis suggested a deterioration of the Noah-MP
simulated LAI when irrigation was activated over the cropland area. We interpreted this correlation deterioration by the
absence of specific information about the crop phenology in the model parameterization. In practice, information about the
specific crop type is not available and the rooting depth is the sole parameter controlling water uptake from the soil layers.
Additionally, information on sowing and harvest periods are not included in the current version of Noah-MP, while irrigated
areas are defined based on a global dataset (Salmon et al., 2013) which can suffer accuracy limitations. Indeed, the absence of
annual dynamic information on irrigated fields, the unknown yearly variability of the crop types and the impact of the
meteorological conditions in the stakeholders decision process (i.e., sowing) make the simulation of Noah-MP prone to LAI
peak shifts, as compared to observations, when irrigation is simulated. This results in a significant performance deterioration
(often worse than LAI simulation not including irrigation which are mainly driven by seasonality, see Figure 7). By contrast,
irrigation modeling helps in reducing the bias of the LAI simulated time series, which, in the cropland area, show a significant
underestimation when irrigation is not considered.

The limitations found in simulating LAI and vegetation by Noah-MP even when irrigation was simulated could potentially be
overcome by assimilating Sentinel-1 $\sigma^0$ data. To explore this potential, we compared the LAI from both model runs, and from
PROBA-V, with the observed Sentinel-1 $\sigma^0$ CR (VH/VV), which should provide information about the vegetation dynamics
(Vreugdenhil et al. 2018; Vreugdenhil et al. 2020). We found that the correlation between $\sigma^0$ CR and LAI from PROBA-V was
much higher than that between $\sigma^0$ CR and the simulated LAI by Noah-MP (see Figure 7) suggesting that Sentinel-1 $\sigma^0$ DA
could help in correcting poor LAI model simulations. Additionally, a higher correlation was found between the $\sigma^0$ VV





observations and the simulated SSM when irrigation was turned on than in the absence of irrigation, suggesting that the assimilation of $\sigma^0$ VV could improve SSM where irrigation is poorly or not modeled.

Finally, by-weekly accumulated irrigation estimates in Figure 7 agree well with real irrigation applications, suggesting that the
large-scale LSM irrigation scheme is helpful for intensively irrigated areas. On the other hand, the poor soil and crop parameterization along with other unknown parameters related to the irrigation management (e.g., the farmers can apply more water than actually needed) can cause large biases in these irrigation simulations. Again, ingestion of radar backscatter data could correct for unmodelled processes. More specifically, Sentinel-1 $\sigma^0$ could correct: (i) for the magnitude and timing of the irrigation simulations; and (ii) for Noah-MP irrigation predictions over not irrigated regions.

**4.2 WCM backscatter simulation**

The purpose of the presented WCM observation operator calibration and evaluation was to optimize the parameters for the future assimilation of the Sentinel-1 $\sigma^0$ VV and VH into Noah-MP. Such an optimization would ideally minimize the long-term bias between the simulated and observed $\sigma^0$ signals. This can be achieved by calibrating the observation operator with long-term observed $\sigma^0$ prior to data assimilation, but in this process, it is crucial to avoid potential error cross-correlation
between model observation predictions and observations. Furthermore, a good observation operator should not already compensate for missing processes in the LSM by accepting effective, but unrealistic, optimized parameters, because it would then lose its physically-based ability to accurately convert misfits between observations and simulations to LSM updates during the data assimilation. In this line, we considered two different experiments: a *Natural* run and an *Irrigation* run, as well as two cost functions, a Bayesian solution *J* and a *KGE* solution which resulted in four calibration experiments for each polarization
(eight calibration experiments).

The calibration experiments using simulations from the *Natural* run as input showed a limited performance and provided presumably bad vegetation parameter estimates which resulted in unrealistic peaks in the simulated $\sigma^0$ during the summer, when driven by higher modelled LAI during this period. The inclusion of the irrigation within Noah-MP was very beneficial for all the calibration experiments helping in reducing the bias and increasing the correlation with Sentinel-1 $\sigma^0$ as well as
removing the anomalous $\sigma^0$ increase during warm periods especially for the *KGE*-based calibration. This corroborates our initial hypothesis that, over intensively irrigated areas, the simulation of irrigation is a mandatory task for an optimal calibration of the WCM. Irrigation modeling, even if only done approximately and perhaps with inaccurate timing, reduces obvious land surface (soil moisture, vegetation) bias and avoids that the WCM needs to compensate for this bias.

Our results show overall higher performance in terms of KGE and Pearson-R scores for the *KGE*-based calibration, whereas
the long-term bias was better reduced for the *J*-based calibration, which is beneficial in anticipation of future DA. This is because in the *J* cost function i) a target accuracy term which takes into account also the Sentinel-1 observations error is present; and, ii) a parameter deviation penalty based on the prior parameters constraints is used, which avoid parameters to largely deviate to their prior values.



In terms of polarization, we found $\sigma^0$ VH simulations much more sensitive to the inclusion of the irrigation (vs non inclusion)
in Noah-MP, suggesting that observed $\sigma^0$ VH might also contain much more information about irrigation (via the influence of
the vegetation change due to irrigation) than that contained in $\sigma^0$ VV which is normally used for SSM retrieval (Vreugdenhil
et al. 2020). We believe that the cause of this is related to a comparatively larger $\sigma^0$ of vegetation with respect to that of the
soil when the crops are well developed. This was also corroborated by the better agreement between CR and LAI from PROBA-
V in one of the study sites mentioned above. Despite this, further investigations are required to confirm this hypothesis and
DA will certainly help to test this aspect.

## 5 CONCLUSIONS

With the specific focus on intensively irrigated land, the main objective of this work was to define the optimal calibration of
the WCM as observation operator for the future ingestion of Sentinel-1 backscatter into the Noah-MP LSM via DA. In this
context, we additionally aimed at: 1) unveiling strengths and limitations of irrigation simulation in LSMs from the perspective
of a calibrating the WCM; 2) identifying the potential irrigation-related information contained in the Sentinel-1 $\sigma^0$ observations
to improve soil moisture and vegetation states as well as irrigation estimates in a calibrated DA system.

To reach these objectives we coupled the Noah-MP with a sprinkler irrigation scheme within LIS and performed two different
simulation experiments, one with and one without irrigation (i.e., *Natural* and *Irrigation* runs). Moreover, we coupled a WCM
with Noah-MP and tested different calibration options to prepare for optimal, future, assimilation of $\sigma^0$ VV and VH to update
both soil moisture and vegetation states.

 The main conclusions drawn from our evaluation are as follows:

- Over highly irrigated areas, the simulation of irrigation in LSMs helps to provide better soil moisture and vegetation
  simulations which can be used with benefit as input for the WCM calibration. However, the performance of the
  irrigation simulations is limited by the simplistic model parameterization of this human process and the necessity to
consider realistic and updated land cover information (e.g., crop types). This results in poor simulations of the
  irrigation timing and quantities as well as vegetation dynamics.
- The Sentinel-1 $\sigma^0$ observations contain useful information about SSM and vegetation over highly irrigated areas. This
  information can be exploited to overcome LSM deficiencies in simulating soil moisture and vegetation over highly
  irrigated regions, e.g., when irrigation is unmodeled, or poorly modeled because of uncertainties due to crop types,
irrigation timing, and farmer agricultural practices. In particular, there is a high chance that the assimilation of
  Sentinel-1 $\sigma^0$ can help in correcting LAI dynamics.
- The optimal assimilation of Sentinel-1 $\sigma^0$ into a LSM must rely upon a well calibrated WCM as observation operator
  to provide unbiased $\sigma^0$ simulations with a minimal chance of having error cross-correlations between model and
  observations, while ensuring a realistic operator controllability or realistic connection between observed signals and
land surface state variables. We demonstrated that calibrating the WCM with inclusion of irrigation modeling



consistently led to a better agreement with Sentinel-1 $\sigma^0$. The modeling of irrigation in the LSM simulations, even if not done optimally, avoids that the WCM calibration compensates for LSM biases.

- We demonstrated that the WCM calibration with a Bayesian cost function, including a prior parameter constraint, provides the optimal WCM parameters, able to generate the lowest bias in the $\sigma^0$ simulations for both VV and VH. Although slightly higher correlations are obtained when using a *KGE* cost function, unbiased estimates are particularly beneficial for DA as this minimizes the chance of potential error cross-correlation between model estimates and observations.

This study improves the understanding of the LSM limitations in simulating irrigation and highlights the information content in Sentinel-1 $\sigma^0$ data. A natural follow up of this study is the assimilation of $\sigma^0$ observations within Noah-MP which should enforce our tested evidence and provide new insights for a more realistic description of the water and carbon cycles over irrigated areas.

**ACKNOWLEDGEMENT**

The authors would like to thank the European Space Agency (ESA) for the funding support as part of the IRRIGATION+ project (contract n. 4000129870/20/I-NB). For details please visit https://esairrigationplus.org/. Additionally, the authors want to acknowledge the Vlaams Supercomputer Centrum (VSC) High Performance Computing (HPC) for providing the computational resources needed to realize this study (https://www.vscentrum.be/). Alexander Gruber has received funding from the Research Foundation Flanders (FWO-1224320N and FWO-1530019N).

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



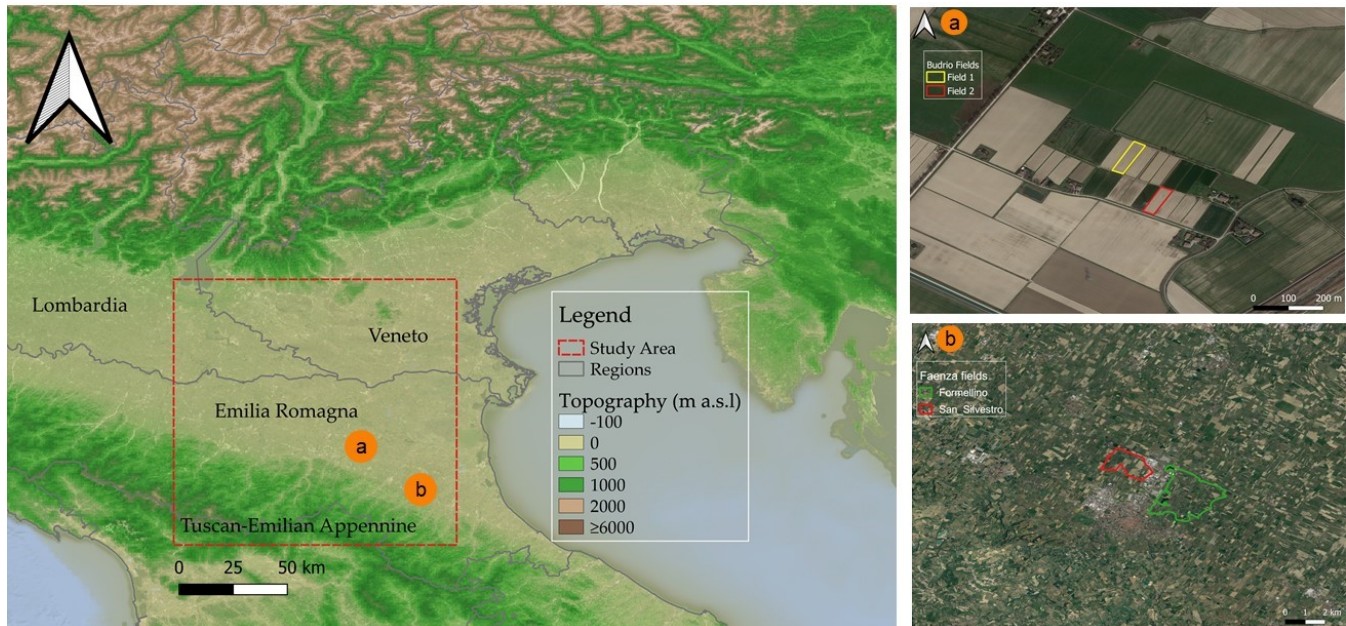

**Figure 1: The study area and the two test sites of (a) Budrio and (b) Formellino. Data on the topography are obtained from ETOPO1 Arc-Minute Global Relief Model (Amante & Eakins, 2009). Map data ©2015 Google.**

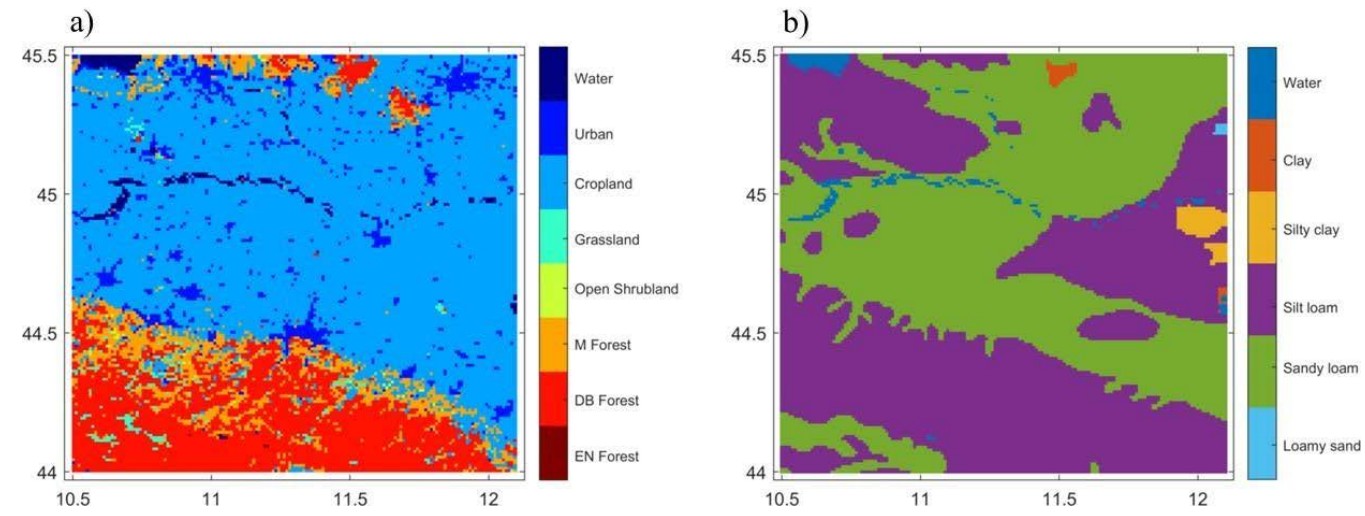


**Figure 2: Re-gridded and reclassified input data used in the LIS framework: a) the PROBA-V LC map; and b) the HWSD soil texture map.**



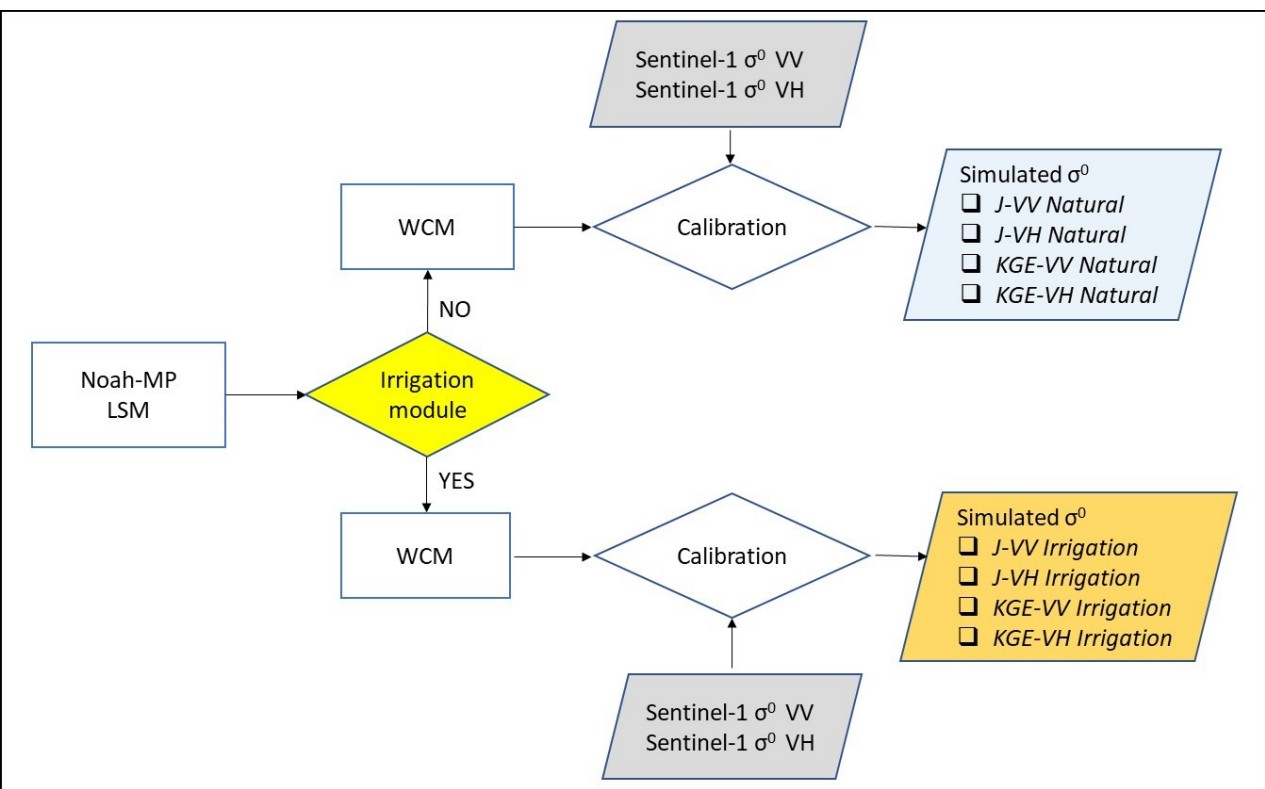

**Figure 3**: **Flow chart of the experimental setup used in this study to calibrate the WCM σ0 signal. A *Natural* and an *Irrigation* experimental line was performed coupling either Noah-MP *Natural* or *Irrigation* simulations with the WCM. For each experimental line σ$^0$ simulations are driven by the Sentinel-1 signal using two different cost functions (*J* and *KGE*) in order to provide eight different calibration experiments.**



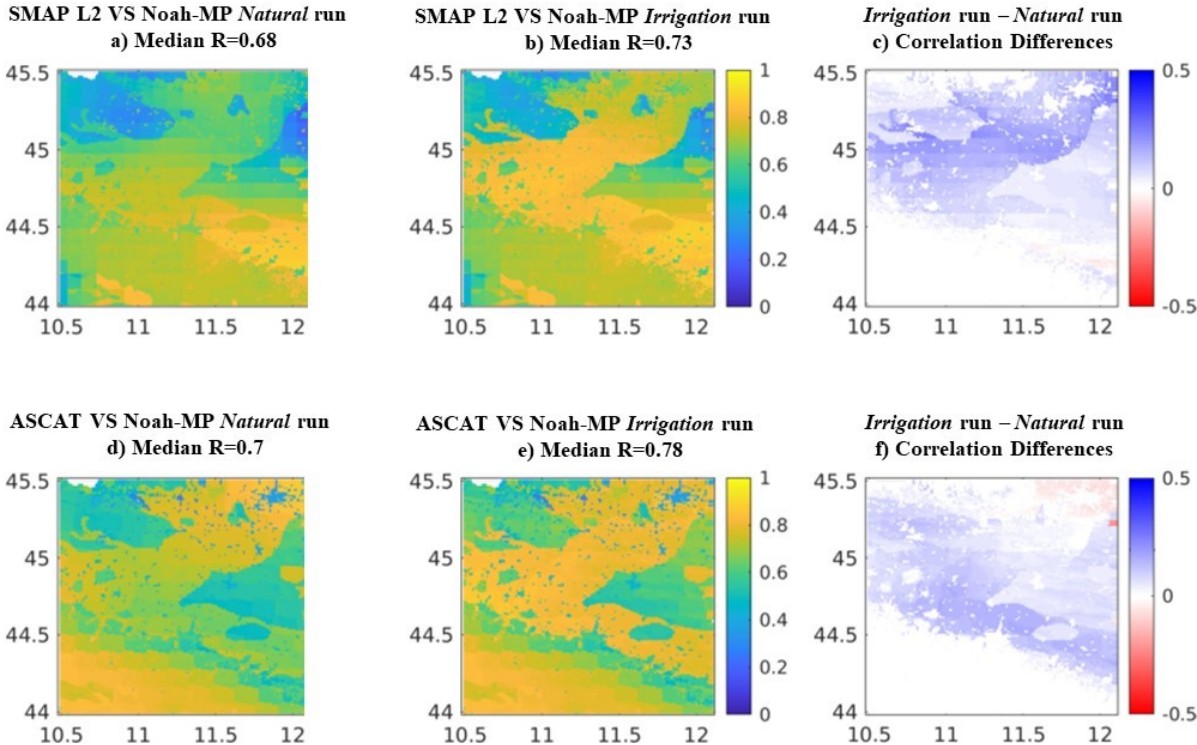

**Figure 4: Maps of temporal Pearson-R between bi-weekly values of SSM from Noah MP and satellite retrievals: a)** *Natural* **run and SMAP L2; b)** *Irrigation* **run and SMAP L2; d)** *Natural* **run and ASCAT; e)** *Irrigation* **run and ASCAT. Maps of the Pearson-R differences display the grid-based difference between: c) map b and map a; f) map e and map d. The reference period is April 2015- December 2019.**



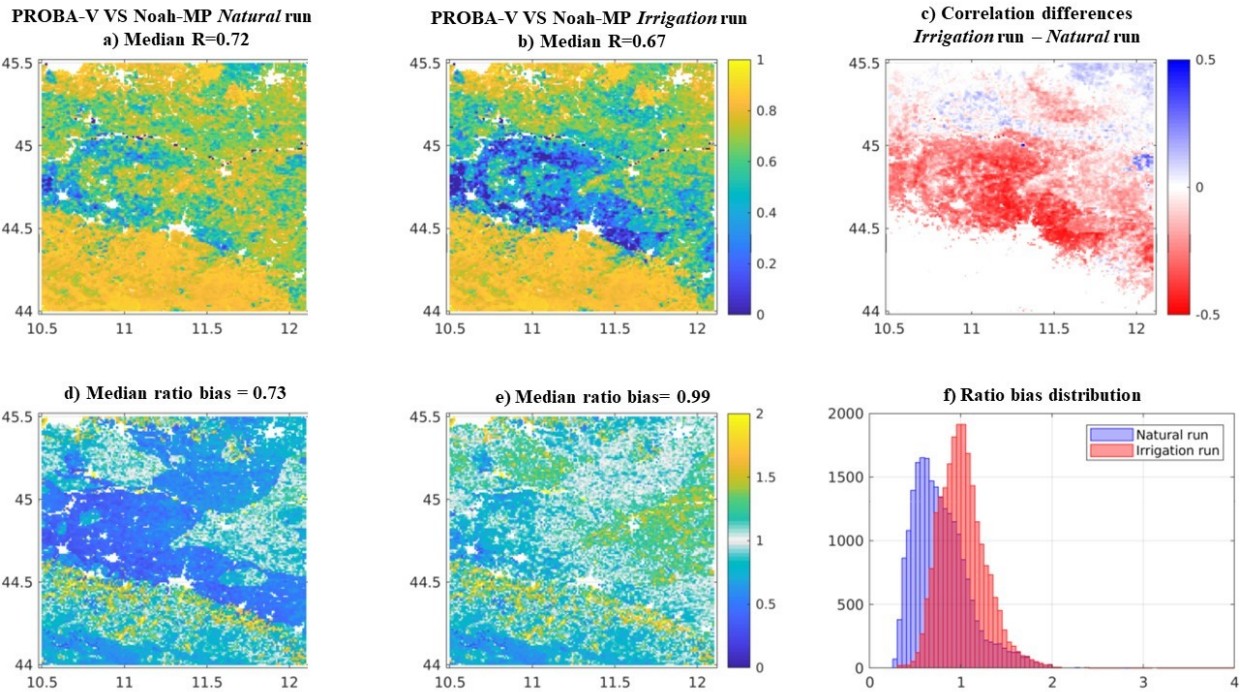

**Figure 5: Maps of temporal Pearson-R between dekadal values of LAI from PROBA-V LAI and Noah-MP LAI: a)** *Natural* **run; b)** *Irrigation* **run. Map of Pearson-R differences between: c) map b and map a. Map of ratio bias of LAI from PROBA-V and Noah-MP: d)** *Natural* **run; e)** *Irrigation* **run. Additional histogram distributions from: f) map d and map e. The reference period is January 2015-October 2019.**

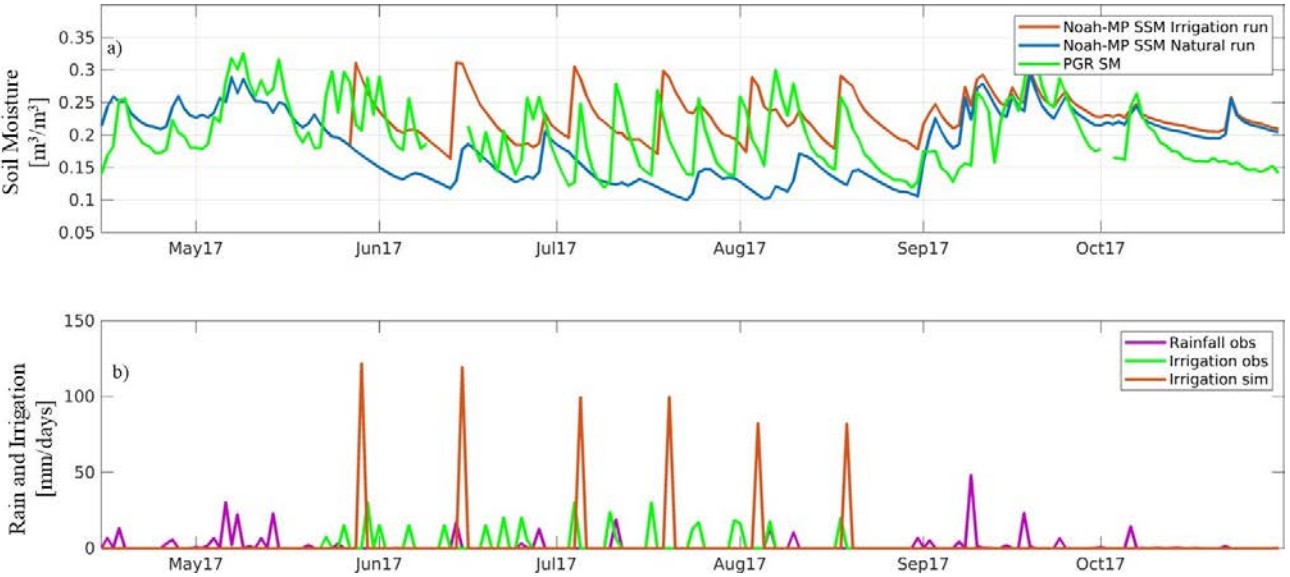

**Figure 6: Evaluation of SSM over the Budrio field 2, with (green) in situ PGR SM data, (light blue) SSM from Noah-MP** *Natural* **and (orange) SSM from Noah-MP** *Irrigation***. Additional information is provided in the bottom plot: b) observed irrigation (green), simulated irrigation (orange) and observed rainfall (magenta) in mm/day**



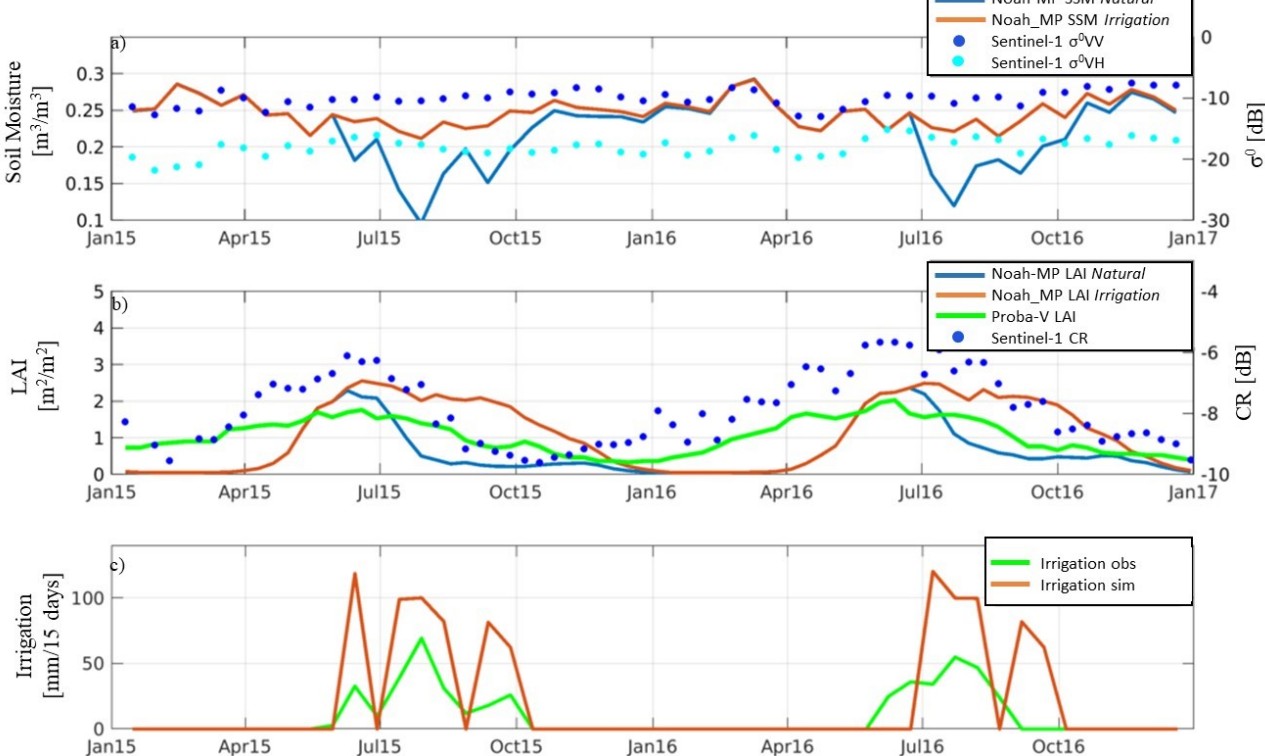

**Figure 7: Sentinel-1 σ⁰ VV and VH data for the Budrio field 1 test site compared with Noah-MP SSM, for a)** *Natural* **and** *Irrigation*
**runs. Sentinel-1 CR (VH/VV) compared with PROBA-V LAI and Noah-MP LAI for b)** *Natural* **and** *Irrigation* **runs. Also shown are:**
**c) observed irrigation (in green) and simulated irrigation from Noah-MP (in orange).**



**Figure 8: Split violin distributions of the calibrated parameters over the entire study area for the eight calibration experiments. For both the *Natural* (blue) and *Irrigation* (orange) experiments, the distributions are shown for the A, B, C, and D parameters, (a, b, c, d) using the *KGE* objective function for VV polarization, (e, f, g, h) *J* objective function for VV polarization, (i, j, k, l) *KGE* objective function for VH polarization, and (m, n, o, p) *J* objective function for VH polarization. Note that the areas under the histograms on both left and right sides of the violins are automatically scaled for optimizing the visualization.**





**Figure 9: Split violin distributions of (a, b, c, d) Pearson-R, (e, f, g, h) KGE and (i, j, k, l) bias calculated between σ⁰ simulations and observations for the validation period, for all the calibration experiments and considering only the cropland areas, using simulations from the *Natural* run (left, green) and the *Irrigation* run (right, violet). The results are shown for VV (first two columns) and VH (right two columns), and alternating for both the calibration with a *J* and *KGE* cost function. Note that the areas under the histograms on both left and right sides of the violins are automatically scaled for optimizing the visualization.**



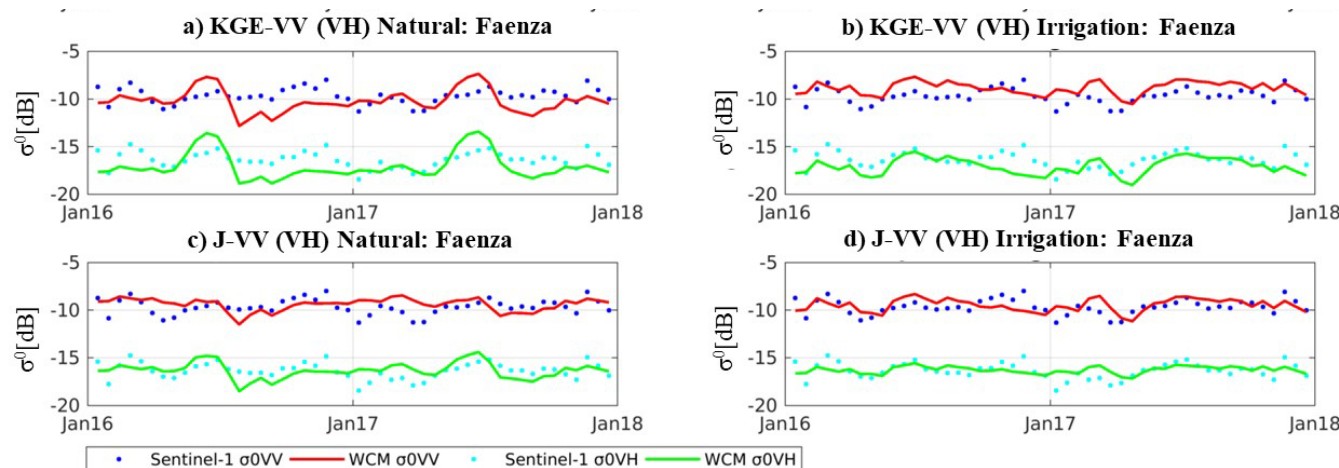


**Figure 10: Comparisons between $\sigma^0$ observations (VV polarization in blue dots and VH polarization light blue dots) and simulations (VV polarization in red and VH polarization in green) in the Faenza San Silvestro field, after calibration with a *KGE* cost function for a) the *Natural* run, b) *Irrigation* run, and after calibration with the *J* cost function for c) the *Natural*, and d) *Irrigation* run.**




**Code and Data availability.**

Data from SMAP can be downloaded at https://nsidc.org/data/SPL2SMP_E/versions/4

Data from ASCAT are available at the website http://hsaf.meteoam.it/

The Sentinel-1 backscatter data processing was done using Google Earth Engine's Python interface and including standard

processing techniques

Data from PROBA-V are available at https://land.copernicus.eu/global/

MERRA-2 data are available at MDISC, managed by the NASA Goddard Earth Sciences (GES) Data and Information Services

Center (DISC, https://disc.gsfc.nasa.gov/datasets?project=MERRA-2)

LIS input and general parameters tables are provided at https://portal.nccs.nasa.gov/lisdata_pub/data/

In situ data are available under request to the original providers.

**Author contributions**: Sara Modanesi designed, coordinated the study and made the analyses. Christian Massari, Gabrielle
De Lannoy and Alexander Gruber designed, coordinated and helped in the data analysis and interpretation. Hans Lievens,
Angelica Tarpanelli and Renato Morbidelli helped in the result interpretation and data processing and collection. All authors

contributed to the editing of the manuscript.

**Competing interests.** The authors declare no competing interests