# Peer review of "Optimizing a backscatter forward operator using Sentinel-1 data over"

_Hydrology and Earth System Sciences, 2021_

## Author Comment (AC1)

**General Comment:**

Modeling irrigation in earth system models is facing different sources of uncertainties and utilizing satellite products via data assimilation could be an effective way to constrain and improve irrigation simulation and its effects on the terrestrial water, carbon, and energy cycles. This study evaluates the irrigation simulation in Noah-MP, identifies the potential of Sentinel-1 observations in containing irrigation signals and discusses the potential of assimilating this observation into Noah-MP in improving irrigation simulation. I found the study interesting and valuable as the exploration of high-resolution remote sensing products in improving model representation of agricultural activities could be valuable in improving modeling of hydrological and carbon cycles under human regulation and in providing info for water management in the future. However, I do think there are some sections need to be improved and clarified, and further discussion is needed regarding revealing the benefits of assimilating the observations into the model. Please see my specific comments below:

> **Reply. We thank the reviewer for having caught the relevance of the study and for the valuable comments. We will improve the manuscript based on the specific comments below.**

**Specific comments:**

**(1) I found the abstract a bit misleading as the study is only exploring the potential of Sentinel-1 sigma-0 observations in containing irrigation signals and providing evaluations in preparation for data assimilation instead of a data assimilation paper. For instance, it is difficult to connect WCM calibration with optimizing Noah-MP by reading only the abstract. I would suggest the authors to reorganize the second paragraph of the abstract to avoid vague statement of the scientific goal and the content of the study.**

> **Reply. We thank the reviewer for this valuable comment. This work indeed does not address the data assimilation of Sentinel-1 itself but the preparation for an optimal data assimilation system which needs an optimal calibration of the observation operator (i.e., the WCM in our case). In this context, optimizing the land modelling system means optimizing a coupled system that includes the Noah-MP LSM and the WCM (used to simulate backscatter predictions). We do agree that the second paragraph needs reorganization to better focus on the content of the study and thus we will improve it in the revised manuscript.**

**(2)L 49-51: I didn't get the logistics using "either…or…". Are the authors trying to say one of the shortcomings of irrigation parameterization in existing studies is not specifying the source water (Ozdogan et al. 2010b, Evans and Zaitchik, 2008), and even if source water partitioning is considered in Nie et. al. (2018), it only includes groundwater irrigation, instead of dividing the source into different parts? Please clarify and rephrase.**

> **Reply. Thanks for this comment. The meaning of this sentence is exactly the one highlighted by the reviewer. In the LIS public source code, it is possible to simulate irrigation without specifying the source of water or, as in Nie et al. (2018), to extract water from a simplified modelled groundwater such as in Noah-MP. In our specific case, this second option is not optimal considering that in the Po river valley the majority of irrigation water comes from surface water (i.e., Po river). We will edit the text in the revised manuscript to better clarify this concept.**

**(3) Why assuming a spatially distributed parameter sets (A, B, C, D) instead of a uniform distribution? I wonder whether the authors analyze the spatial pattern of the parameter distribution**

and is there any obvious patterns or stratifications of the parameters relating to soil types, climate types, or anything else? Showing this would help audience understand better the meaning of those parameters and relate that to why Natural and Irrigation runs lead to different calibration performance.

Reply. Thank you for this comment. Note that the final objective of the WCM calibration is to reduce the long-term bias between Sentinel-1 and the simulated backscatter signal grid cell by grid cell, for future data assimilation experiments. Following previous works (Lievens et al. 2017; De Lannoy et al. 2013; De Lannoy et al., 2014) we implemented a grid-based calibration instead of using a uniform distribution in order to take into account the spatial differences between observed and modelled backscatter caused by the model parameterization of soil and vegetation, and specific features in the observed footprint. This will be better clarified in the revised manuscript.

We also analyzed the spatial pattern of the parameters and found a certain connection with land uses and soil texture as shown in Figure 2 of the manuscript. An example of parameter maps is reported in Figure R1 for the J-VV Natural and J-VV Irrigation experiments. Generally, the activation of the irrigation scheme seems to reduce the dependency of the vegetation parameters to the soil texture (the reviewer can refer for instance to the low A-values on the triangle structure at the eastern side of the study area in the Natural experiment --Figure R1 a --, which do not appear in the Irrigation experiment --Figure R1 e). On the other hand, the C and D parameters, which refer to the bare soil backscatter, seem to be more dependent on the soil texture in the Irrigation experiment (Figures R1g and R1h). Here, the big central triangle structure is highlighted as compared to the Natural experiment (lower C values and higher D values). In this area the sandy-loamy soil allows more irrigation water as compared to the less permeable silty-loam texture of the eastern triangle structure. We agree that showing maps of parameters can result in a better understanding of the different calibration experiments. We will point this out in the revised manuscript and add a specific paragraph for discussing this issue.

[Figure]

Figure R1. Maps of: a) A parameter; b) B parameter; c) C parameter; d) D parameter for the *J-VV Natural* calibration experiment. Maps of: e) A parameter; f) B parameter; g) C parameter; h) D parameter for the *J-VV Irrigation* calibration experiment.

(4) Irrigation affects SSM and LAI, leading to different parameter distribution in WCM calibration process. However, there are mixed results when evaluating against observed SSM and LAI products. For instance, Irrigation run provides improved estimation of LAI magnitude, while degradation in LAI temporal variability. I wonder whether the authors can calibrate the WCM model using the observed SSM and LAI product, and compare the difference in parameter distribution. How does that look like and what could be the uncertainties in retrieving these parameters purely depending on Noah-MP or depending on observations? In other words, could the authors elaborate the

discussion on the uncertainty of the calibrated parameters and for example quantify how capturing the LAI magnitude vs. LAI temporal variation would contribute to the calibration of WCM?

Reply. Thanks for pointing this out. The optimal calibration of WCM is indeed a challenging task and can be implemented by following different strategies depending on the final target. In this particular case, our goal is to build an observation operator tuned on model inputs, for future data assimilation experiments so, theoretically, model-based SSM and LAI should be used. On the other hand, while using observed 1km-SSM is practically unfeasible (as these data are normally not available at this resolution but from backscatter, e.g., from S1) the use of PROBA-V-based LAI can be a valuable alternative to model-based LAI. This would allow us to overcome the problem of the mismatch of the temporal dynamic between the true and modelled vegetation caused by the model parameterization of irrigation. However, a preliminary assessment shows very different LAI time series from different satellite sensors and missing data due to cloud cover, and after all, it is not LAI per se but the water in the vegetation which is needed for backscatter simulation. Imposing the use of an observed LAI product can therefore also introduce additional bias in the backscatter model simulations, undermining the optimality of the data assimilation experiments. We will add some text to discuss this aspect in the revised version of the manuscript.

(5) L349-352: I didn't quite understand the rationale of "minimizing the impact of the irrigation signal already contained in sigma-0 observations". Why activating irrigation can minimize this impact? And if the impact is minimized, how you can utilize the irrigation related info in data assimilation if detectable in sigma-0 observations? Please clarify.

Reply. Thanks for this comment. As discussed in the introduction at lines 115-119 if the WCM inputs (SSM and LAI from Noah-MP) miss crucial processes such as irrigation, then the WCM calibration (tuned on Sentinel-1, which theoretically contains the irrigation signal) will compensate for this bias providing correlated errors between the WCM and observations in the future data assimilation experiments. Activating the irrigation scheme reduces this risk as SSM and LAI inputs from Noah-MP contain the irrigation signal and the calibration system will not be "forced" to correct for unmodelled processes. We will make this point clearer in the revised version of the manuscript.

(6) The simulation is performed at 0.01 deg while part of the evaluation is conducted at field level, the area of which is much smaller than the model space. Could the authors discuss the uncertainties that might be associated with this evaluation due to the scale mismatch?

Reply. Thanks for this comment. We think that the reviewer has highlighted a crucial point, especially when modelling human activities such as irrigation. One of the most critical aspects of irrigation validation is given by the lack of irrigation benchmark data (Foster et al., 2020). In this specific case, we decided to not exclude the Budrio test site considering the reliability of the data over the fields. The second aspect is that we realized evaluations at different scales: 1) regional (the entire study area); 2) small-district (Faenza test site); and 3) plot scale (Budrio fields). Indeed, while the Budrio test site is composed by plots of about 0.4 hectares, the analysis over the Faenza test site (see Figure 10) refers to an area of 270 ha which is comparable with the model estimates. Finally, it has to be noted that we have selected an intensively irrigated area. Maps such as the Global Rainfed Irrigation Paddy Areas (GRIPC; Salmon et al., 2015) confirm that the Po river valley is almost entirely irrigated, thus reducing the risk to find non-irrigated fields within the 1-km LIS grid. We are aware that there are limitations in our approach but we think that 0.01° spatial resolution is a good compromise between analysis on a regional, small-district and plot scale. We will add a small section to better clarify this point in the revised version of the manuscript.

(7) Figure 7 (a): could the authors elaborate a bit why simulated soil moisture can be directly compared to the VV and VH data, and what might be the difference between the VV and VH data

regarding the detection of soil moisture? What might be the reason for negative correlation between simulated SSM and VH for both Natural and Irrigation runs?

Reply. Thanks for highlighting this aspect. VV polarization of radar backscatter is more strictly linked to SSM information than VH signal though both of them include information on soil properties (see Gruber et al., 2013; Wagner et al., 2013; Bauer-Marschallinger et al., 2018). For instance, Baghdadi et al. (2017) found that the soil's contribution to total backscattering coefficient is lower in VH than in VV because VH is more sensitive to vegetation cover and that, the use of VH alone to retrieve soil moisture, is suboptimal when vegetation cover is well developed. In this context, the cross-polarization backscatter (i.e., HV and VH signals) was found to be well related to vegetation in previous studies (i.e., Ferrazzoli et al., 1992; Macelloni et al., 2001). Based on that, we compared soil moisture directly with VV (and to VH to understand the soil contribution to it) and CR with LAI. This also provides insights about the potential of VV and VH to update soil moisture and vegetation. We will clarify this aspect in the revised version of the manuscript.

(8) L462-463: It is encouraging to see that the CR has a strong relation to the vegetation signal and could be potentially used to correct the simulated vegetation phenology. However, I was confused how exactly this calibration framework could be introduced to Noah-MP DA? Are you suggesting approximating CR to LAI and directly assimilating CR into the model? Or if you are using calibrated parameter to assimilate into Noah-MP, how does the CR information is going to be ingested? I would suggest the authors to clarify and provide more in detail how the current study is connected to assimilating sigma-0 observations into Noah-MP as I found it is unclear throughout the text.

Reply. Thank you for this comment. Firstly, we plan to couple the WCM with Noah-MP in LIS. The WCM is our observation operator, which means that the parameters obtained from the calibration will be used to simulate observation predictions ($\sigma^\circ$-VV and $\sigma^\circ$-VH) in LIS using the WCM. We plan to implement different experiments (i.e., assimilate VV to update SSM, VH to update LAI, or both VV and VH to update SSM and LAI simultaneously for instance). Another option, indeed, could be to directly assimilate the CR to update LAI. This is a future step which will be discussed in the following work focusing on ingestion of Sentinel-1 backscatter to improve irrigation quantification. We will introduce this clarification in the Discussion section of the revised manuscript.

(9) L486-491: Why a more uniform distributed C and D is more realistic? If so, why the calibrated parameters are designed to be spatially distributed?

Reply. Thanks for this comment. We agree with the reviewer that this aspect needs more clarification. We meant that the Natural experiment provided parameters values more squeezed towards the lower/upper defined boundaries. This means that the parameters are not well constrained and optimized while the Irrigation experiment shows a higher spread in the parameters. In this context, as suggested by the reviewer we will also add a spatial analysis of the results showing that the C and D parameters are more influenced by the soil moisture dynamics in the Irrigation experiments. We will clarify this aspect in the text and we will remove the term "uniform".

(10) What is the benefit of assimilating sigma-0 observations instead of directly assimilating LAI or SSM products? I think the authors should discuss and highlight the benefit of assimilating sigma-0 observations in both introduction section and results section.

Reply. Thanks for this comment. We partly discussed this methodological choice in the introduction section at lines 96 to 104. As reported in De Lannoy et al. (2016), a critical aspect in directly assimilating SSM retrievals is that potentially inconsistent ancillary data are used in the assimilation system and in the retrieval algorithm that generates SSM observations. Furthermore, active MW retrievals typically use change detection methods (Wagner et al., 2013; Bauer-Marshallinger et al., 2018) which lack land-specific information. This means that the 'error

**management' within the data assimilation system is theoretically more transparent when assimilating backscatter observations. Using microwave retrievals allows us to have consistent parameters between the LSM and the radiative transfer model (in our case the WCM) and to avoid cross-correlated errors.**

**Technical corrections:**

L57: remove "to" after "from".
**Reply. Thanks for this comment. We will correct the text.**

L102: "focussed" -> "focused".
**Reply. Thanks for this comment. We will correct the text.**

L195: Add a space between and "observations"
**Reply. Thanks for this comment. We will correct the text.**

L528 & L530: "Table 3" should be "Table 2"?
**Reply. Thanks for this comment. We will correct the text.**

**References**

- Bauer-Marschallinger, B., Freeman, V., Cao, S., Paulik, C., Schaufler, S., Stachl, T., Modanesi, S., Massari, C., Ciabatta, L., Brocca, L. and Wagner, W., (2018). Toward Global Soil Moisture Monitoring With Sentinel-1: Harnessing Assets and Overcoming Obstacles. IEEE
- De Lannoy, G.J.M., Reichle, R.H. Assimilation of SMOS brightness temperatures or soil moisture retrievals into a land surface model Hydrol. Earth Syst. Sci., doi:10.5194/hess-2016-414, 2016
- De Lannoy, G.J.M., Reichle, R.H., Vrugt, J.A.: Uncertainty quantification of GEOS-5 L-band radiative transfer model parameters using Bayesian inference and SMOS observations Remote Sens. Environ., 148, pp. 146-157, doi:/10.1016/j.rse.2014.03.030, 2014.
- De Lannoy, G.J.M., Reichle, R., Pauwels, V.: Global calibration of the GEOS-5 L-band microwave radiative transfer model over non-frozen land using SMOS observations, Journal of Hydrometeorology, 14, 765–785, doi:/10.1175/JHM-D-12-092., 2013.
- Ferrazzoli, P.; Paloscia, S.; Pampaloni, P.; Schiavon, G.; Solimini, D.; Coppo, P. Sensitivity of microwave measurements to vegetation biomass and soil moisture content: A case study. IEEE Trans. Geosci. Remote Sens. 1992, 30, 750–756.
- Foster, T.; Mieno, T.; Brozovic, N. Satellite-based monitoring of irrigation water use: assessing measurement errors and their implications for agricultural water management policy. Water Resour. Res. 2020, 56, 10.1029/2020WR028378
- Gruber, A.; Wagner, W.; Hegyiova, A.; Greifeneder, F.; Schlaffer, S. Potential of Sentinel-1 for high-resolution soil moisture monitoring. In 2013 IEEE International Geoscience and Remote Sensing Symposium-IGARSS; IEEE: Melbourne, Australia, 2013; pp. 4030–4033
- Lievens, H., Martens, B., Verhoest, N. E. C., Hahn, S., Reichle, R. H., Miralles, D. G.: Assimilation of global radar backscatter and radiometer brightness temperature observations to improve soil moisture and land evaporation estimates, Remote Sens. Environ., 189, 194–210, doi: 10.1016/j.rse.2016.11.022, 2017
- Macelloni, G.; Paloscia, S.; Pampaloni, P.; Marliani, F.; Gai, M. The relationship between the backscattering coefficient and the biomass of narrow and broad leaf crops. IEEE Trans. Geosci. Remote Sens. 2001, 39, 873–884.
- Salmon, J. M., Friedl, M. A., Frolking, S., Wisser, D.: Global rain-fed, irrigated, and paddy croplands: A new high resolution map derived from remote sensing, crop inventories and climate data, International Journal of Applied Earth Observation and Geoinformation, 38, 321–334, doi:/10.1016/j.jag.2015.01.014 , 2015.
- Wagner, W., Hahn, S., Kidd, R., Melzer, T., Bartalis, Z., Hasenauer, S., Figa-Saldaña, J., de Rosnay, P., Jann, A., Schneider, S., Komma, J., Kubu, G., Brugger, K., Aubrecht, C., Züger, J., Gangkofner, U., Kienberger, S., Brocca,

L., Wang, Y., Blöschl, G., Eitzinger, J., and Steinnocher, K.: The ASCAT soil moisture product: A review of its specifications, validation results, and emerging applications, Meteorol. Z., 22, 5–33, 2013

---

## Author Comment (AC2)

**General Comment:**

Irrigation is the largest human intervention in the water cycle, yet it is poorly represented in the land surface (LSM) and hydrological models. One way to account for irrigation is by assimilating the observations that contain the irrigation signal, such as radar backscatter ($\sigma^0$) or satellite soil moisture retrieval. One important step prior to the assimilation is removing biases between the model and the observation through calibration. In this study, the Noah MP model is coupled with a backscatter observation model (WCM), and Sentinel-1 $\sigma^0$ observations were used to calibrate the model. Furthermore, the impact of activating the irrigation schemes within the Noah MP model, using different backscatter polarization (VV or VH) and cost functions in model calibration, is investigated. I found the study interesting; however, as a calibration study, I expected that the results be more focused on the calibration rather than evaluating the performance of the LSM. I also have some concerns regarding the mixed results obtained regarding the activation of the irrigation module. Please see my comments for detail:

Reply. We would like to thank the reviewer for this general comment and for the interest in the manuscript. We will implement the following specific comments more in depth.

**Specific comments:**

1. **L175: The size of the Budrio test sites is much smaller than the model resolution (almost 1/200 of the model spatial resolution). I do not think it is a good choice for validation purposes.**

Reply. Thanks for this comment. We agree that this is a crucial point, especially when modelling human activities such as irrigation. One of the most critical aspects of irrigation validation is given by the lack of irrigation benchmark data (Foster et al. 2020). In this specific case, we decided to not exclude the Budrio test site considering the reliability of the data over the fields. The second aspect is that we realized evaluations at different scales: 1) regional (the entire study area); 2) small-district (Faenza test site); and 3) plot scale (Budrio fields). Indeed, while the Budrio test site is composed by plots of about 0.4 hectares, the analysis over the Faenza test site (see Figure 10) refers to an area of 270 Ha which is comparable with the model estimates. Finally, it has to be noted that we have selected an intensively irrigated area. Maps such as the Global Rainfed Irrigation Paddy Areas (GRIPC; Salmon et al., 2015) confirm that the Po river valley is almost entirely irrigated with a percentage of about 100% over the 0.01° LIS grid. This reduces the risk of finding non-irrigated fields within the 1km LIS grid. We are aware that there are limitations in our approach but we think that 0.01° spatial resolution is a good compromise between analysis on a regional, small-district and plot scale. We plan to add a small section in the revised manuscript to better clarify this point.

2. **L300-305: It is not indicated which vegetation indicator (VWC, NDVI, or LAI) is finally chosen to represent V1 and V2 in equation 2 and 3. Moreover, it is not clear whether it is assumed that V1=V2 and a unique descriptor is used for both of them or not? According to the rest of the paper, I suppose that LAI is the chosen vegetation indicator, but I think this should be explicitly mentioned here.**

Reply. Thanks for this comment. We agree with the reviewer. This point is not explicitly mentioned. We assumed V1=V2, represented by the dynamically simulated LAI vegetation descriptor. We will add this explanation in the text of the revised manuscript.

3.      **Section 2.4: As mentioned in L95, assimilating the SSM or VWC retrieval instead of MW brightness temperature or σ0 can be problematic due to inconsistent ancillary data used in their production, and $\sigma^0$ is a better choice for assimilation. However, as is explained in section 2.4, assimilation of $\sigma^0$ requires the NOAH MP model to be coupled with a WCM model to simulate $\sigma^0$. In turn, the WCM model has many empirical parameters and simplifying assumptions such as not accounting for scattering interactions between ground and vegetation and assuming a linear relation between soil $\sigma^0$ and the SSM that can increase the uncertainty of the estimated $\sigma^0$. Given this, can the authors clarify why assimilating $\sigma^0$ is a better choice relative to the assimilation of SSM and VWC?**

> **Reply. Thanks for this comment. A valuable reply to this comment is contained in previous papers we have cited in the introduction section at lines 94-101. As reported in De Lannoy et al. (2016), a critical aspect in directly assimilating SSM retrievals is that potentially inconsistent ancillary data are used in the assimilation system and in the retrieval algorithm that generates SSM observations. Furthermore, active MW retrievals typically use change detection methods (Wagner et al., 2013; Bauer-Marshallinger et al., 2018) which lack land-specific information. This means that the 'error management' within the data assimilation system is theoretically more transparent when assimilating backscatter observations. Using microwave retrievals allows us to have consistent parameters between the LSM and the radiative transfer model (in our case the WCM) and to avoid cross-correlated errors.**

4.      **L356: Another interesting comparison would be comparing the performance of a third calibration approach which is, deactivating the irrigation scheme and calibrating the model only during the non-irrigated season, with the current approach (activating the irrigation scheme and calibrating for the entire period) during the non-irrigated season. This would also be an interesting comparison for the future study in which $\sigma^0$ will be assimilated to see whether calibrating during the irrigation season with the activated irrigation module is beneficial for the ultimate goal of irrigation quantification or not.**

> **Reply. Thanks for this comment. We tested the additional calibration option of removing the irrigated periods (i.e., summer) when the irrigation scheme is deactivated. We found a not reliable saturation of the vegetation parameters (especially the A parameter) through the upper boundary over the cropland areas with wrong simulations during the summer, affected by very high summer peaks in the backscatter signal (especially affecting the backscatter VV polarization). In this specific case, the higher contribution to the signal is given by the SSM which is higher during the winter. On the other hand, the backscatter from the vegetation cannot provide a large contribution as the LAI signal is low during the winter. This prompts the vegetation parameters to be pushed towards their upper boundaries. An example of parameters maps for the KGE-VV Natural experiment, removing the irrigation period, is shown in Figure R2 below. We also added an example time series plot (Figure R2e) where we compare the Sentinel-1 and the WCM signals over a pixels in the cropland area where the A reaches a value of 0.4 [-] (upper boundary). The blue time series, representing the WCM signal, clearly shows the anomalous peaks during the summer period.**

[Figure]

**Figure R2. Maps of: a) A parameter; b) B parameter; c) C parameter; d) D parameter for the *KGE-VV Natural* calibration experiment, realized removing summer irrigation period. e) Time series of simulated and observed (Sentinel-1) backscatter for an example grid cell.**

We also observed this phenomenon in the Natural experiment considering the entire study period (summer+winter) but at a smaller scale and for an opposite reason as discussed at lines 480-486.

Thanks for the suggestion, we will briefly include this thought and our findings in the discussion of the revised manuscript.

5.      **L405-416: I am not convinced that the improvement in the simulation of SSM by Noah MP is due to the activation of the irrigation module for two reasons:**
- **In most of the previous studies, it is shown that coarse-scale MW products are not able to detect irrigation signals at the plot scale (e.g., Brocca et al., 2019, Zaussinger et al., 2018, Dari et al., 2020) unless there is intensive flood irrigation over a large area such as California central valley in which fields are flooded, and the water level is sustained throughout the irrigation season (Lawston et al., 2017). Moreover, to have a fair comparison, other metrics such as RMSE or bias should also be reported alongside the Pearson correlation (R).**
- **The deterioration in LAI simulation when irrigation is activated makes more sense to me as the spatial resolution of the Proba-V LAI product is 1km, and the possibility of detecting the irrigation signal is higher relative to the coarse-scale SSM products.**

      **Please comment on this.**

Reply. Thanks for this comment. Po river Valley is one of the largest and most intense irrigated areas in Europe so it is very likely that the contribution of the irrigation in the coarse-scale SSM is visible especially at biweekly temporal scale (i.e., the majority of the land within the satellite footprint is irrigated).

On the other hand, for the second point, we agree with the Reviewer. The higher spatial resolution of the Proba-V results in finer spatial details about crop state due to plot-specific agricultural practices, the unknown yearly variability of the crop types and the impact of the meteorological conditions in the stakeholders decision process as we already explained at lines 568-580 of the discussion section which are difficult to represent when irrigation is activated (for the simplified model parameterization of the irrigation process). Furthermore, temporal dynamics of LAI are clearly more sensitive to root zone soil moisture which might be more difficult to simulate than SSM during the irrigation season due to very likely larger impact of the soil texture and

**transpiration processes along with the high frequency of the wetting and drying caused by irrigation events.**

**Additional discussion will be provided on these aspects in the revised version of the manuscript where also other metrics will be used as suggested by the reviewer.**

**6.** **L435-446: I think the part of this significant difference between simulated and observed irrigation and missing the irrigation events is related to the spatial mismatch between the model and the test site, as it is mentioned in the first comment. Please comment on this.**

**Reply. Thanks for this comment. The reviewer can refer to the reply to comment (1)**

**References**

- Bauer-Marschallinger, B., Freeman, V., Cao, S., Paulik, C., Schaufler, S., Stachl, T., Modanesi, S., Massari, C., Ciabatta, L., Brocca, L. and Wagner, W., (2018a). Toward Global Soil Moisture Monitoring With Sentinel-1: Harnessing Assets and Overcoming Obstacles. IEEE
- De Lannoy, G.J.M., Reichle, R.H. Assimilation of SMOS brightness temperatures or soil moisture retrievals into a land surface model Hydrol. Earth Syst. Sci., doi:10.5194/hess-2016-414, 2016
- Foster, T.; Mieno, T.; Brozovic, N. Satellite-based monitoring of irrigation water use: assessing measurement errors and their implications for agricultural water management policy. Water Resour. Res. 2020, 56, 10.1029/2020WR028378
- Salmon, J. M., Friedl, M. A., Frolking, S., Wisser, D.: Global rain-fed, irrigated, and paddy croplands: A new high resolution map derived from remote sensing, crop inventories and climate data, International Journal of Applied Earth Observation and Geoinformation, 38, 321–334, doi:/10.1016/j.jag.2015.01.014 , 2015.
- Wagner, W., Hahn, S., Kidd, R., Melzer, T., Bartalis, Z., Hasenauer, S., Figa-Saldaña, J., de Rosnay, P., Jann, A., Schneider, S., Komma, J., Kubu, G., Brugger, K., Aubrecht, C., Züger, J., Gangkofner, U., Kienberger, S., Brocca, L., Wang, Y., Blöschl, G., Eitzinger, J., and Steinnocher, K.: The ASCAT soil moisture product: A review of its specifications, validation results, and emerging applications, Meteorol. Z., 22, 5–33, 2013

---

## Author Response (AR1)

**Editor: Evaluation**

Dear Authors,

Thank you for your detailed responses to the two reviews of your paper. As you can see, the reviews are both generally quite positive, but both reviewers have requested further discussion and clarification of the approach and methods. Please implement the suggested changes thoroughly and the manuscript will be further reviewed by myself and the referees.

I look forward to reading your revised manuscript.

Sincerely,

Louise Slater

**Reply:**

We would like to thank the Associate Editor and all the Reviewers for the good feedbacks and valuable comments. We have now carefully updated our manuscript based on the comments below. As an additional note, during the revision process we found a bug in the script we used for extracting Sentinel-1 data, which resulted in a slight displacement (~1 km) of the Sentinel-1 dataset grid compared to the LIS Noah-MP grid. We reproduced the calibration experiments and updated Figures 7, 8, 9 (now 10) and 10 (now 11), the ones affected by our correction. We apologize for this mistake. Note that given the low variability of the Noah-MP SSM and LAI simulations at the 1 km scale no substantial differences were introduced.

Please note:

✔   **All modifications are shown in red color both in the text and in this document.**

**I. Reviewer#1 Evaluation and comments**

**General Comment:**

Modeling irrigation in earth system models is facing different sources of uncertainties and utilizing satellite products via data assimilation could be an effective way to constrain and improve irrigation simulation and its effects on the terrestrial water, carbon, and energy cycles. This study evaluates the irrigation simulation in Noah-MP, identifies the potential of Sentinel-1 observations in containing irrigation signals and discusses the potential of assimilating this observation into Noah-MP in improving irrigation simulation. I found the study interesting and valuable as the exploration of high-resolution remote sensing products in improving model representation of agricultural activities could be valuable in improving modeling of hydrological and carbon cycles under human regulation and in providing info for water management in the future. However, I do think there are some sections need to be improved and clarified, and further discussion is needed regarding revealing the benefits of assimilating the observations into the model. Please see my specific comments below:

> **Reply. We thank the reviewer for articulating the relevance of the study and for the valuable comments. We will improve the manuscript based on the specific comments below.**

**General Comment:**

**(1) I found the abstract a bit misleading as the study is only exploring the potential of Sentinel-1 sigma-0 observations in containing irrigation signals and providing evaluations in preparation for data assimilation**

instead of a data assimilation paper. For instance, it is difficult to connect WCM calibration with optimizing Noah-MP by reading only the abstract. I would suggest the authors to reorganize the second paragraph of the abstract to avoid vague statement of the scientific goal and the content of the study.

Reply. We thank the reviewer for this valuable comment. This work indeed does not address the data assimilation of Sentinel-1 itself (which will be the subject of further research), but the preparation for an optimal data assimilation system which requires calibration of the observation operator (i.e., the WCM in our case). In this context, optimizing the land modelling system means optimizing a coupled system that includes the Noah-MP LSM and the WCM (used to simulate backscatter predictions). We do agree that the second paragraph needs reorganization and we have changed it as follows (lines 21-38 of the track-changes version of the manuscript)

This work represents the first and necessary step towards building a reliable LSM data assimilation system which, in future analysis, will investigate the potential of high-resolution radar backscatter observations from Sentinel-1 to improve irrigation quantification. Specifically, the aim of this study is to couple the Noah-MP LSM running within the NASA Land Information System (LIS), with a backscatter observation operator for simulating unbiased backscatter predictions over irrigated lands. In this context, we first tested how well modelled Surface Soil Moisture (SSM) and vegetation estimates, with or without irrigation simulation, are able to capture the signal of aggregated 1-km Sentinel-1 backscatter observations over the Po river Valley, an important agricultural area in Northern Italy. Next, Sentinel-1 backscatter observations, together with simulated SSM and LAI, were used to optimize a Water Cloud Model (WCM) which will represent the observation operator in future data assimilation experiments. The WCM was calibrated with and without an irrigation scheme in Noah-MP, and considering two different cost functions. Results demonstrate that using an irrigation scheme provides a better calibration of the WCM, even if the simulated irrigation estimates are inaccurate. The Bayesian optimization is shown to result in the best unbiased calibrated system, with minimal chance of having error cross-correlations between the model and observations. Our time series analysis further confirms that Sentinel-1 is able to track the impact of human activities on the water cycle, highlighting its potential to improve irrigation, soil moisture, and vegetation estimates via future data assimilation.

(2) L 49-51: I didn't get the logistics using "either…or…". Are the authors trying to say one of the shortcomings of irrigation parameterization in existing studies is not specifying the source water (Ozdogan et al. 2010b, Evans and Zaitchik, 2008), and even if source water partitioning is considered in Nie et. al. (2018), it only includes groundwater irrigation, instead of dividing the source into different parts? Please clarify and rephrase.

Reply. Thanks for this comment. The meaning of this sentence is exactly the one highlighted by the reviewer. In the LIS public source code it is possible to simulate irrigation without specifying the source of water or, as in Nie et al. (2018), to extract water from a simplified modelled groundwater such as in Noah-MP. In our specific case, this second option is not optimal considering that in the Po river valley the majority of irrigation water comes from surface water (i.e., the Po river). We have edited the text as follows at lines 53-58 (track-changes version of the manuscript):

In earlier studies, attempts to simulate irrigation in LSMs have relied on different parameterizations of well-known irrigation systems (like sprinkler, flood, and drip systems; Ozdogan et al., 2010b; Evans and Zaitchik, 2008), making simplifying assumptions. For instance, in Ozdogan et al. (2010b) irrigation water is not withdrawn from a source (such as a river) but instead added as fictitious rainfall. In contrast, Nie et al. (2018) accounted for source water partitioning, albeit only partially, by considering groundwater irrigation.

(3) Why assuming a spatially distributed parameter sets (A, B, C, D) instead of a uniform distribution? I wonder whether the authors analyze the spatial pattern of the parameter distribution and is there any obvious patterns or stratifications of the parameters relating to soil types, climate types, or anything else?

**Showing this would help audience understand better the meaning of those parameters and relate that to why Natural and Irrigation runs lead to different calibration performance.**

**Reply. Thank you for this comment. Note that the final objective of the WCM calibration is to reduce the long-term bias between Sentinel-1 and the simulated backscatter signal grid cell by grid cell, for future data assimilation experiments. Following previous works (Lievens et al. 2017; De Lannoy et al. 2013; De Lannoy et al., 2014) we implemented a grid-based calibration instead of using a uniform distribution in order to take into account the spatial differences between observed and modelled backscatter caused by the model parameterization of soil and vegetation, and specific features in the observed footprint. This aspect has been clarified in the revised manuscript at lines 387-391 (track-changes version):**

The A, B, C, and D parameters of the WCM (see section 2.4) were fitted separately to Sentinel-1 $\sigma^0$ VV and VH observations, during the period January 2017 - December 2019. Following previous literature (Lievens et al., 2017b; De Lannoy et al., 2014; De Lannoy et al., 2013), we performed a grid cell-based calibration to account for the spatial variability in the simulated and observed $\sigma^0$ signals that stems from specific features within the observed footprints as well as from the soil and vegetation parameterization of Noah-MP.

**We also analyzed the spatial pattern of the parameters and found a certain connection with land uses and soil texture as shown in Figure 2 of the manuscript. An example of parameter maps is reported in Figure R1 for the *J-VV Natural* and *J-VV Irrigation* experiments. Generally, the activation of the irrigation scheme seems to reduce the dependency of the vegetation parameters to the soil texture (see, for instance, the low A-values within the triangle structure at the eastern side of the study area in the *Natural* experiment --Figure R1 a --, which do not appear in the *Irrigation* experiment --Figure R1 e). On the other hand, the C and D parameters, which refer to the bare soil backscatter, seem to be more dependent on the soil texture in the *Irrigation* experiment (Figures R1g and R1h). Here, the big central triangle structure is highlighted as compared to the *Natural* experiment (lower C values and higher D values). In this area, the sandy-loamy soil allows more irrigation water than the less permeable silty-loam texture of the eastern triangle structure.**

[Figure]

**Figure R1. Maps of: a) A parameter; b) B parameter; c) C parameter; d) D parameter for the *J-VV Natural* calibration experiment. Maps of: e) A parameter; f) B parameter; g) C parameter; h) D parameter for the *J-VV Irrigation* calibration experiment.**

**We agree that showing maps of parameters can result in a better understanding of the different calibration experiments and we have added figure R1 in the main text (while all the maps of calibrated parameters -Figures S4 and S5- were added in the Supplementary material). Additionally, a specific paragraph to discuss this aspect has been reported at lines 536-544 of Section 3.3 (track-changes version of the manuscript):**

Figure 9 shows the spatial pattern of the parameters over the study area to better understand the differences between the *Natural* and *Irrigation* calibration runs. We found a connection between the WCM parameters distribution and model parameters, in particular with the HWSD soil texture map (shown in Figure 2). For

both the *J-VV Natural* and *J-VV Irrigation* experiments, the activation of the irrigation scheme reduces the dependency of the vegetation-related parameters A and B on soil texture (Figures 9a-b for the *J-VV Natural* and Figures 9e-f for the *J-VV Irrigation* experiment). This is also shown in the parameter maps of the *KGE* calibration experiments (Figure S5 in the Supplementary material). Additionally, the activation of the irrigation scheme, more realistically, shifts the soil texture dependency towards the soil parameters C and D (Figures 9g and 9h), highlighting another important reason for simulating irrigation.

**(4) Irrigation affects SSM and LAI, leading to different parameter distribution in WCM calibration process. However, there are mixed results when evaluating against observed SSM and LAI products. For instance, Irrigation run provides improved estimation of LAI magnitude, while degradation in LAI temporal variability. I wonder whether the authors can calibrate the WCM model using the observed SSM and LAI product, and compare the difference in parameter distribution. How does that look like and what could be the uncertainties in retrieving these parameters purely depending on Noah-MP or depending on observations? In other words, could the authors elaborate the discussion on the uncertainty of the calibrated parameters and for example quantify how capturing the LAI magnitude vs. LAI temporal variation would contribute to the calibration of WCM?**

> **Reply. Thanks for pointing this out. The optimal calibration of WCM is indeed a challenging task and can be implemented by following different strategies depending on the final target. In this particular case, our goal is to build an observation operator tuned on model inputs to support future data assimilation experiments. Therefore, model-based SSM and LAI should be used in principle. On the other hand, while using observed 1km-SSM is practically unfeasible (as these data are normally not available at this resolution but from backscatter, e.g., from S1) the use of PROBA-V-based LAI can be a valuable alternative to model-based LAI. This would indeed allow us to overcome the problem of the mismatch of the temporal dynamic between the true and modelled vegetation caused by the model parameterization of irrigation. However, a preliminary assessment shows very different LAI time series from different satellite sensors, which are also missing a lot of data due to cloud cover. Imposing the use of an observed LAI product can therefore also introduce additional bias in the backscatter model simulations, undermining the optimality of the data assimilation experiments. We have added some text to discuss this aspect in the revised version of the manuscript at lines 664-670 (track-changes version of the manuscript):**

>> One way to avoid parameters compensation for erroneous LSM input into the WCM would be to use observed time series of e.g. LAI. However, LAI products from different sensors have different biases themselves which can add bias to the $\sigma^0$ simulations, and more importantly, replacing simulated LAI or SSM with external datasets would undermine the possibility of updating these variables in the future assimilation system. Based on that, we performed the WCM calibration considering SSM and LAI model input from two different experiments: a *Natural* run and an *Irrigation* run, as well as two cost functions, a Bayesian solution *J* and a *KGE* solution which resulted in four calibration experiments for each polarization (i.e., eight calibration experiments in total).

**(5) L349-352: I didn't quite understand the rationale of "minimizing the impact of the irrigation signal already contained in sigma-0 observations". Why activating irrigation can minimize this impact? And if the impact is minimized, how you can utilize the irrigation related info in data assimilation if detectable in sigma-0 observations? Please clarify.**

> **Reply. Thanks for this comment. As discussed in the introduction at lines 125-131 (track-changes version), if the WCM inputs (SSM and LAI from Noah-MP) miss crucial processes such as irrigation, then the WCM calibration (tuned on Sentinel-1, which theoretically contains the irrigation signal) will compensate for this bias providing correlated errors between the WCM and observations in the future data assimilation experiments. Activating the irrigation scheme reduces this risk, as SSM and LAI inputs from Noah-MP contain the irrigation signal and the calibration system will not be "forced" to correct for unmodelled processes. We rephrased the sentence and added additional specifications in the revised version of the manuscript at lines 370-376 (track-changes version):**

An optimal DA system requires long-term unbiased σ⁰ simulations (with respect to the assimilated observations). The risk, over an intensively irrigated area, is that an unmodelled irrigation signal would manifest itself as a predominant bias in the σ⁰ simulations. The calibration would then inadvertently correct for this supposed bias (i.e., the irrigation signal), thus preventing the DA system from propagating the missing irrigation signal from the observations into the model. Even though existing irrigation schemes are evidently unrealistic and inaccurate, we conjecture that using such a scheme when calibrating the WCM will more likely yield optimal WCM parameters than when neglecting irrigation.

**(6) The simulation is performed at 0.01 deg while part of the evaluation is conducted at field level, the area of which is much smaller than the model space. Could the authors discuss the uncertainties that might be associated with this evaluation due to the scale mismatch?**

**Reply. Thanks for this comment. We think that the reviewer has highlighted a crucial point, especially when modelling human activities such as irrigation. One of the most critical aspects of irrigation validation is given by the lack of irrigation benchmark data (Foster et al., 2020). In this specific case, we decided to not exclude the Budrio test site notwithstanding the scale difference between the size of test site and the model grid size. The second aspect is that we realized evaluations at different scales: 1) regional (the entire study area); 2) small-district (Faenza test site); and 3) plot scale (Budrio fields). Indeed, while the Budrio test site is composed by plots of about 0.4 hectares, the analysis over the Faenza test site (see Figure 10) refers to an area of 270 ha which is comparable with the model estimates. Furthermore, it has to be noted that we have selected an intensively irrigated area. Maps such as the Global Rainfed Irrigation Paddy Areas (GRIPC; Salmon et al., 2015) confirm that the Po river valley is almost entirely irrigated, thus reducing the risk to find non-irrigated fields within the 1-km LIS grid. We are aware that additional limitations in our approach could be related to crop/field specific timing and magnitude of irrigation but at the moment, large-scale high resolution input (i.e., dynamic crop maps) are not available for LSM simulations and we think that 0.01° spatial resolution is a good compromise between analysis on a regional, small-district and plot scale. We have specified the difference between Budrio fields and the Faenza test site spatial scale analysis in the study area description at lines 185-186 and 200-201 (see section 2.1):**

- For an analysis at plot scale we selected the Budrio test site (Figure 1a), an experimental farm managed by the CER authority which includes two plots of 0.39-0.49 ha.
- The second test site (Figure 1b) is located around the city of Faenza (hereafter Faenza test site) and has a total extent of 1051 ha, consisting of two fields which allow an analysis at the small-district spatial scale.

**Additionally, the following specification was added at lines 405-412 (track-changes version):**

The evaluation was carried out on both the regional scale (i.e., over the entire study area) and on the two selected sites, Faenza (small-district scale) and Budrio (plot scale), where irrigation data were available. Considering the lack of benchmark data for irrigation evaluation (Foster et al., 2020) we decided to use in situ data for the small Budrio fields spatial scale (i.e., 0.45-049 Ha) even though model simulations are made at a much coarser resolution (i.e., ~1 km). We are aware that differences in spatial scale can increase the uncertainty of our evaluation, but 0.01° LSM spatial resolution is still a good compromise for an analysis at regional, small-district and plot scale. Additionally, limitations are partly reduced by the low chance of including non-irrigated fields within the 1 km LIS grid cells within the Po River Valley, as the latter is almost entirely irrigated (Salmon et al., 2015).

**(7) Figure 7 (a): could the authors elaborate a bit why simulated soil moisture can be directly compared to the VV and VH data, and what might be the difference between the VV and VH data regarding the detection of soil moisture? What might be the reason for negative correlation between simulated SSM and VH for both Natural and Irrigation runs?**

**Reply. Thanks for highlighting this aspect. VV polarization of radar backscatter is more strictly linked to SSM information than VH signal though both of them include information on soil properties (see Gruber et al., 2013; Wagner et al., 2013; Bauer-Marschallinger et al., 2018). For instance, Baghdadi et al. (2017) found that the soil's contribution to total backscattering coefficient is lower in VH than in VV**

because VH is more sensitive to vegetation cover and that the use of VH alone to retrieve soil moisture is suboptimal when vegetation cover is well developed. In this context, the cross-polarization backscatter (i.e., HV and VH signals) was found to be well related to vegetation in previous studies (i.e., Ferrazzoli et al., 1992; Macelloni et al., 2001). Based on that, we compared soil moisture directly with VV (and to VH to understand the soil contribution to it) and CR with LAI.  This also provides insights about the potential of VV and VH to update soil moisture and vegetation. To better address this aspect we added the following text at lines 490-492 (section 3.2):

Although the $\sigma^0$ VV is generally used to retrieve SSM (Wagner et al., 2013; Gruber et al., 2013; Bauer-Marschallinger et al., 2018), data at both polarizations were analyzed in order to understand the soil contribution contained in the two signals.

Additional text was also added at lines 498-500 (section 3.2):

On the other hand, $\sigma^0$ VH seems to provide poor performances also when irrigation is simulated, with a Pearson-R value equal to 0.06, confirming findings by Baghdadi et al. (2017) which highlighted how the use of VH alone to retrieve SSM is suboptimal when vegetation cover is well developed.

**(8) L462-463: It is encouraging to see that the CR has a strong relation to the vegetation signal and could be potentially used to correct the simulated vegetation phenology. However, I was confused how exactly this calibration framework could be introduced to Noah-MP DA? Are you suggesting approximating CR to LAI and directly assimilating CR into the model? Or if you are using calibrated parameter to assimilate into Noah-MP, how does the CR information is going to be ingested? I would suggest the authors to clarify and provide more in detail how the current study is connected to assimilating sigma-0 observations into Noah-MP as I found it is unclear throughout the text.**

**Reply. Thank you for this comment. Firstly, we plan to couple the WCM with Noah-MP in LIS. The WCM is our observation operator, which means that the parameters obtained from the calibration will be used to simulate observation predictions ($\sigma^0$-VV and $\sigma^0$-VH) in LIS using the WCM. Considering that the update of a state variable in a data assimilation system depends on its covariance with the backscatter observations, we plan to implement different experiments (i.e., assimilate VV to update SSM, VH to update LAI, or both VV and VH to update SSM and LAI simultaneously for instance). Another option, indeed, could be to directly assimilate the CR to update LAI. This is a future step which will be discussed in the following work focusing on ingestion of Sentinel-1 backscatter to improve irrigation quantification. We have introduced the following clarification in the Discussion section 4.1 at lines 645-648:**

On the other hand, considering the low correlation between the VH signal and SSM in presence of vegetation (Baghdadi et al. 2017), and its close relation with vegetation (Ferrazzoli et al., 1992; Macelloni et al., 2001), future data assimilation experiments will investigate the contribution of VH and CR in improving LAI predictions and irrigation quantification.

**(9) L486-491: Why a more uniform distributed C and D is more realistic? If so, why the calibrated parameters are designed to be spatially distributed?**

**Reply. Thanks for this comment. We agree with the reviewer that this aspect needs more clarification. We meant that the *Natural* experiment provided parameter values more squeezed towards the lower/upper defined boundaries. This means that the parameters are not well constrained and optimized while the *Irrigation* experiment shows a higher spread in the parameters. In this context, as suggested by the reviewer we have also added a spatial analysis of the results (the Reviewer can refer to the reply to comment 3), showing that the C and D parameters are more influenced by the soil moisture dynamics in the *Irrigation* experiments. We have removed the term "uniform" at lines 529-535 of section 3.3:**

The C and D parameter distributions feature a better sensitivity to soil moisture dynamics using the *Irrigation* run input data, which is the expected behaviour considering that they describe the $\sigma^0_{soil}$. This is true especially when using the *J* cost function (see parameters distributions for the *J-VV Natural* and for the *J-VV Irrigation* experiments in Figures 8g and 8h), which results in more spread in the calibrated C and D distributions for the *Irrigation* simulations (especially in VV polarization), whereas the mode of the

C and D parameter distributions for the *Natural* experiments is more shifted to the upper and lower boundaries, respectively.

**(10) What is the benefit of assimilating sigma-0 observations instead of directly assimilating LAI or SSM products? I think the authors should discuss and highlight the benefit of assimilating sigma-0 observations in both introduction section and results section.**

**Reply. Thanks for this comment. We partly discussed this methodological choice in the introduction section. As reported in De Lannoy et al. (2016), a critical aspect in directly assimilating SSM retrievals is that potentially inconsistent ancillary data are used in the assimilation system and in the retrieval algorithm that generates SSM observations. Furthermore, active MW retrievals typically use change detection methods (Wagner et al., 2013; Bauer-Marshallinger et al., 2018a) which lack land-specific information. This means that the 'error management' within the data assimilation system is theoretically more transparent when assimilating backscatter observations. Using microwave observations allows us to have consistent parameters between the LSM and the radiative transfer model (in our case the WCM) and to avoid cross-correlated errors between model states and corresponding geophysical satellite retrievals. Furthermore, top-class SM retrieval algorithms for S1 are still under development (for instance, the 1 km Sentinel-1 SSM available at the Copernicus Global Land Service website Bauer-Marshallinger et al. (2018) does not include yet a correction for vegetation), so relying upon backscatter observations might be advantageous.**

**We added the following text in the introduction section at lines 102-109:**

The assimilation of MW RS observations in LSMs often involves retrieval assimilation. However, assimilating retrievals (i.e., SSM or vegetation optical depth rather than MW brightness temperature or $\sigma^0$ measurements) can be problematic as the retrievals may have been produced with ancillary data that are inconsistent with those used in the LSM (De Lannoy et al., 2016). This is particularly true for passive MW retrievals while active MW retrievals generally rely on change detection methods that lack land-specific ancillary information altogether. An alternative approach, which we follow in this study, is to directly assimilate MW observations and equip the LSM with an observation operator that links land surface variables of interest (e.g., soil moisture and vegetation) with RS data. This allows us to obtain consistent parameters and to reduce the chance of cross-correlated errors between model states and corresponding geophysical satellite retrievals.

**Technical corrections:**

L57: remove "to" after "from".
    **Reply. Thanks for this comment. We have corrected the text.**

L102: "focussed" -> "focused".
    **Reply. Thanks for this comment. We have corrected the text at line 114.**

L195: Add a space between and "observations"
    **Reply. Thanks for this comment. We have corrected the text.**

L528 & L530: "Table 3" should be "Table 2"?
    **Reply. Thanks for this comment. We have corrected the text at line 581.**

**General Comment:**

**Irrigation is the largest human intervention in the water cycle, yet it is poorly represented in the land surface (LSM) and hydrological models. One way to account for irrigation is by assimilating the observations that contain the irrigation signal, such as radar backscatter ($\sigma^0$) or satellite soil moisture retrieval. One important step prior to the assimilation is removing biases between the model and the observation through calibration. In this study, the Noah MP model is coupled with a backscatter observation model (WCM), and Sentinel-1 $\sigma^0$ observations were used to calibrate the model. Furthermore, the impact of activating the irrigation schemes within the Noah MP model, using different backscatter polarization (VV or VH) and cost functions in model calibration, is investigated. I found the study interesting; however, as a calibration study, I expected that the results be more focused on the calibration rather than evaluating the performance of the LSM. I also have some concerns regarding the mixed results obtained regarding the activation of the irrigation module. Please see my comments for detail:**

> **Reply. We would like to thank the reviewer for this general comment and for the interest in the manuscript. We will implement the following specific comments more in depth.**

**Specific comments:**

**(1)     L175: The size of the Budrio test sites is much smaller than the model resolution (almost 1/200 of the model spatial resolution). I do not think it is a good choice for validation purposes.**

> **Reply. Thanks for this comment. We agree that this is a crucial point, especially when modelling human activities such as irrigation. One of the most critical aspects of irrigation validation is given by the lack of irrigation benchmark data (Foster et al. 2020). In this specific case, we decided to not exclude the Budrio test site considering the reliability of the data over the fields. The second aspect is that we realized evaluations at different scales: 1) regional (the entire study area); 2) small-district (Faenza test site); and 3) plot scale (Budrio fields). Indeed, while the Budrio test site is composed by plots of about 0.4 hectares, the analysis over the Faenza test site (see Figure 10) refers to an area of 270 ha which is comparable with the model estimates. Furthermore, it has to be noted that we have selected an intensively irrigated area. Maps such as the Global Rainfed Irrigation Paddy Areas (GRIPC; Salmon et al., 2015) confirm that the Po river valley is almost entirely irrigated, thus reducing the risk to find non-irrigated fields within the 1-km LIS grid. We are aware that additional limitations in our approach could be related to crop/field specific timing and magnitude of irrigation but at the moment, large-scale high-resolution input (i.e., dynamic crop maps) are not available for LSM simulations and we think that 0.01° spatial resolution is a good compromise between analysis on a regional, small-district and plot scale. We have specified the difference between Budrio fields and the Faenza test site spatial scale analysis in the study area description at lines 185-186 and 200-201 (see section 2.1):**

> - For an analysis at plot scale we selected the Budrio test site (Figure 1a), an experimental farm managed by the CER authority which includes two plots of 0.39-0.49 ha.
> - The second test site (Figure 1b) is located around the city of Faenza (hereafter Faenza test site) and has a total extent of 1051 ha, consisting of two fields which allow an analysis at the small-district spatial scale.

> **Additionally, the following specification was added at lines 405-412 (track-changes version):**

> The evaluation was carried out on both the regional scale (i.e., over the entire study area) and on the two selected sites, Faenza (small-district scale) and Budrio (plot scale), where irrigation data were available. Considering the lack of benchmark data for irrigation evaluation (Foster et al., 2020) we decided to use in situ data for the small Budrio fields spatial scale (i.e., 0.45-049 Ha) even though model simulations are made at a much coarser resolution (i.e., ~1 km). We are aware that differences in spatial scale can increase the uncertainty of our evaluation, but 0.01° LSM spatial resolution is still a good compromise for an

analysis at regional, small-district and plot scale. Additionally, limitations are partly reduced by the low chance of including non-irrigated fields within the 1 km LIS grid cells within the Po River Valley, as the latter is almost entirely irrigated (Salmon et al., 2015).

**(1)    L300-305: It is not indicated which vegetation indicator (VWC, NDVI, or LAI) is finally chosen to represent V1 and V2 in equation 2 and 3. Moreover, it is not clear whether it is assumed that V1=V2 and a unique descriptor is used for both of them or not? According to the rest of the paper, I suppose that LAI is the chosen vegetation indicator, but I think this should be explicitly mentioned here.**

**Reply. Thanks for this comment. We agree with the reviewer. This point is not explicitly mentioned. We have added the following explanation in the text at lines 321-323 of Section 2.4:**

Following previous studies (see Lievens et al, 2017b; Baghdadi et al. 2017; Li and Wang, 2018) we assumed $V_1=V_2$ represented by the dynamically simulated LAI vegetation descriptor.

**(2)    Section 2.4: As mentioned in L95, assimilating the SSM or VWC retrieval instead of MW brightness temperature or σ0 can be problematic due to inconsistent ancillary data used in their production, and $\sigma^0$ is a better choice for assimilation. However, as is explained in section 2.4, assimilation of $\sigma^0$ requires the NOAH MP model to be coupled with a WCM model to simulate $\sigma^0$. In turn, the WCM model has many empirical parameters and simplifying assumptions such as not accounting for scattering interactions between ground and vegetation and assuming a linear relation between soil $\sigma^0$ and the SSM that can increase the uncertainty of the estimated $\sigma^0$. Given this, can the authors clarify why assimilating $\sigma^0$ is a better choice relative to the assimilation of SSM and VWC?**

**Reply. Thanks for this comment. We partly discussed this methodological choice in the introduction section. As reported in De Lannoy et al. (2016), a critical aspect in directly assimilating SSM (or VWC) retrievals is that potentially inconsistent ancillary data are used in the assimilation system and in the retrieval algorithm that generates SSM observations. Furthermore, active MW retrievals typically use change detection methods (Wagner et al., 2013; Bauer-Marshallinger et al., 2018a) which lack land-specific information. This means that the 'error management' within the data assimilation system is theoretically more transparent when assimilating backscatter observations. Using microwave observations allows us to have consistent parameters between the LSM and the radiative transfer model (in our case the WCM) and to avoid cross-correlated errors between model states and corresponding geophysical satellite retrievals. Furthermore, top-class SM retrieval algorithms for S1 are still under development (for instance, the 1 km Sentinel-1 SSM available at the Copernicus Global Land Service website Bauer-Marshallinger et al. (2018) does not include yet a correction for vegetation), so relying upon backscatter observations might be advantageous.**

**We added the following text in the introduction section at lines 102-109:**

The assimilation of MW RS observations in LSMs often involves retrieval assimilation. However, assimilating retrievals (i.e., SSM or vegetation optical depth rather than MW brightness temperature or $\sigma^0$ measurements) can be problematic as the retrievals may have been produced with ancillary data that are inconsistent with those used in the LSM (De Lannoy et al. 2016). This is particularly true for passive MW retrievals while active MW retrievals generally rely on change detection methods that lack land-specific ancillary information altogether. An alternative approach, which we follow in this study, is to directly assimilate MW observations and equip the LSM with an observation operator that links land surface variables of interest (e.g., soil moisture and vegetation) with RS data. This allows us to obtain consistent parameters and to reduce the chance of cross-correlated errors between model states and corresponding geophysical satellite retrievals.

**(3)    L356: Another interesting comparison would be comparing the performance of a third calibration approach which is, deactivating the irrigation scheme and calibrating the model only during the non-irrigated season, with the current approach (activating the irrigation scheme and calibrating for the entire period) during the non-irrigated season. This would also be an interesting comparison for the future study**

in which σ⁰ will be assimilated to see whether calibrating during the irrigation season with the activated irrigation module is beneficial for the ultimate goal of irrigation quantification or not.

> **Reply. Thanks for this comment. We tested the additional calibration option of removing the irrigated periods (i.e., summer) when the irrigation scheme is deactivated. We noticed a saturation of the vegetation parameters (especially the A parameter) through the upper boundary over the cropland areas resulting in awkward simulations during the summer, affected by very high peaks in the backscatter signal (especially affecting the backscatter VV polarization). This can be explained by the higher contribution to the signal by the SSM (which is higher during the winter) and the relatively lower LAI. This prompts the vegetation parameters to be pushed towards their upper boundaries. An example of parameters maps for the *KGE-VV Natural* experiment, removing the irrigation period, is shown in Figure R2 below. Figure R2e below compares the Sentinel-1 and the WCM signals over a pixel in the cropland area where A reaches a value of 0.4 [-] (upper boundary). The blue time series, representing the WCM signal, clearly shows the anomalous peaks during the summer period.**

[Figure]

> **Figure R2. Maps of: a) A parameter; b) B parameter; c) C parameter; d) D parameter for the *KGE-VV Natural* calibration experiment, realized removing summer irrigation period. e) Time series of simulated and observed (Sentinel-1) backscatter for an example grid cell.**

**(4)    L405-416: I am not convinced that the improvement in the simulation of SSM by Noah MP is due to the activation of the irrigation module for two reasons:**

**●    In most of the previous studies, it is shown that coarse-scale MW products are not able to detect irrigation signals at the plot scale (e.g., Brocca et al., 2019, Zaussinger et al., 2018, Dari et al., 2020) unless there is intensive flood irrigation over a large area such as California central valley in which fields are flooded, and the water level is sustained throughout the irrigation season (Lawston et al., 2017). Moreover, to have a fair comparison, other metrics such as RMSE or bias should also be reported alongside the Pearson correlation (R).**

**●    The deterioration in LAI simulation when irrigation is activated makes more sense to me as the spatial resolution of the Proba-V LAI product is 1km, and the possibility of detecting the irrigation signal is higher relative to the coarse-scale SSM products.**

**Please comment on this.**

> **Reply. Thanks for this comment. The work by Zaussinger et al. (2018) and Dari et al. (2020) refer to different study areas (CONUS in the first case and Spain for the second study). On the other hand, in the work by Brocca et al. (2019) the Authors selected only two neighbouring pixels (25 × 25 km²) located in the Po river valley, a much smaller area as compared to the one analyzed in this study. The Po river Valley is one of the largest and most intensively irrigated areas in Europe so it is very likely that the contribution of the irrigation in the coarse-scale SSM is visible especially at biweekly temporal scale (i.e., the majority of the land within the satellite footprint is irrigated). We have calculated the RMSE**

as an additional metric for the SSM evaluation rescaling the satellite products based on the long-term mean and standard deviation of the modelled data. Results are shown in Figure R3 below confirming that the activation of the Irrigation scheme provides an improvement in performances also in terms of RMSE for both the satellite products. For instance, figure R3c shows a median RMSE=0.039 when comparing ASCAT and Noah-MP SSM *Natural* runs, while Figure R3d, which compares the ASCAT and Noah-MP SSM *Irrigation* runs, displays an improvement in performance with a decrease in the median value of RMSE to 0.034 m³/m³. An additional confirmation that the improvement can be attributed to the Irrigation scheme is given by the large enhancement observed over the entire Po river valley cropland area, but particularly over the central triangle feature which is classified as sandy-loam in the soil-texture input data. We introduced this additional analysis at Section 2.7 (lines 420-422 of the track-changes version of this manuscript):

For SSM, we also computed the Root Mean Square Error (RMSE), calculated considering the original temporal resolution of the satellite products, while for LAI, we also tested the ratio bias, i.e., the ratio between the long-term mean of the simulations and the long-term mean of observations.

An additional discussion about this aspect has been added at lines 450-456 at the Results section 3.1:

Results in Figure 4 were confirmed by analyzing the RMSE between satellite SSM products and Noah-MP simulations for both the *Natural* and *Irrigation* runs, after rescaling them based on their mean and standard deviation, because SSM retrievals and SSM simulations do not have the same units. Results are displayed in Figure S3 of the Supplementary material and show, for both the satellite products, a general reduction in RMSE when compared with the *Irrigation* run. An improvement in performances can be observed over the entire cropland area, in particular over the central triangle feature where sandy-loam soil texture is present and where, consequently, more irrigation is simulated in the model due to the higher permeability of the soil.

And in the discussion section 4.1 at lines 618-620 the following specifications have been added:

For both products, we found large improvements in temporal Pearson-R when irrigation was simulated, confirmed by a decrease in the RMSE values over croplands, suggesting that the activation of irrigation modelling provides more realistic SSM estimates.

[Figure]

**Figure R3. Maps of RMSE between SSM from Noah MP and satellite retrievals: a) *Natural* run and SMAP L2; b) *Irrigation* run and SMAP L2; c) *Natural* run and ASCAT; d) *Irrigation* run and ASCAT.**

For the second point, we agree with the Reviewer. The higher spatial resolution of the Proba-V results in finer spatial details such as crop state due to plot-specific agricultural practices, the unknown yearly variability of the crop types or the impact of the meteorological conditions in the stakeholders decision

**process, as we already explained at lines 622-637. Those features are scarcely reproducible in the model when irrigation is activated (for the simplified parameterization of the irrigation process). Furthermore, temporal dynamics in LAI are clearly more sensitive to root zone soil moisture which might be more difficult to simulate than SSM during the irrigation season due to very likely larger impact of the soil texture and transpiration processes along with the high frequency of the wetting and drying caused by irrigation events. We have added to the already mentioned discussion the following text at lines 631-634 (track-changes version):**

Another important aspect affecting LAI simulations is its sensitivity to root zone soil moisture, which might be more difficult to simulate than SSM during the irrigation season due to larger impacts of the soil texture and transpiration processes along with the high frequency of the wetting and drying phases caused by irrigation events.

**(5)      L435-446: I think the part of this significant difference between simulated and observed irrigation and missing the irrigation events is related to the spatial mismatch between the model and the test site, as it is mentioned in the first comment. Please comment on this.**

**Reply. Thanks for this comment. The reviewer can refer to the reply to comment (1)**

**References**

- Bauer-Marschallinger, B., Freeman, V., Cao, S., Paulik, C., Schaufler, S., Stachl, T., Modanesi, S., Massari, C., Ciabatta, L., Brocca, L. and Wagner, W., (2018). Toward Global Soil Moisture Monitoring With Sentinel-1: Harnessing Assets and Overcoming Obstacles. IEEE
- De Lannoy, G.J.M., Reichle, R.H. Assimilation of SMOS brightness temperatures or soil moisture retrievals into a land surface model Hydrol. Earth Syst. Sci., doi:10.5194/hess-2016-414, 2016
- De Lannoy, G.J.M., Reichle, R.H., Vrugt, J.A.: Uncertainty quantification of GEOS-5 L-band radiative transfer model parameters using Bayesian inference and SMOS observations Remote Sens. Environ., 148, pp. 146-157, doi:/10.1016/j.rse.2014.03.030, 2014.
- De Lannoy, G.J.M., Reichle, R., Pauwels, V.: Global calibration of the GEOS-5 L-band microwave radiative transfer model over non-frozen land using SMOS observations, Journal of Hydrometeorology, 14, 765–785, doi:/10.1175/JHM-D-12-092., 2013.
- Ferrazzoli, P.; Paloscia, S.; Pampaloni, P.; Schiavon, G.; Solimini, D.; Coppo, P. Sensitivity of microwave measurements to vegetation biomass and soil moisture content: A case study. IEEE Trans. Geosci. Remote Sens. 1992, 30, 750–756.
- Foster, T.; Mieno, T.; Brozovic, N. Satellite-based monitoring of irrigation water use: assessing measurement errors and their implications for agricultural water management policy. Water Resour. Res. 2020, 56, 10.1029/2020WR028378
- Gruber, A.; Wagner, W.; Hegyiova, A.; Greifeneder, F.; Schlaffer, S. Potential of Sentinel-1 for high-resolution soil moisture monitoring. In 2013 IEEE International Geoscience and Remote Sensing Symposium-IGARSS; IEEE: Melbourne, Australia, 2013; pp. 4030–4033
- Lievens, H., Martens, B., Verhoest, N. E. C., Hahn, S., Reichle, R. H., Miralles, D. G.: Assimilation of global radar backscatter and radiometer brightness temperature observations to improve soil moisture and land evaporation estimates, Remote Sens. Environ., 189, 194–210, doi: 10.1016/j.rse.2016.11.022, 2017
- Macelloni, G.; Paloscia, S.; Pampaloni, P.; Marliani, F.; Gai, M. The relationship between the backscattering coefficient and the biomass of narrow and broad leaf crops. IEEE Trans. Geosci. Remote Sens. 2001, 39, 873–884.
- Salmon, J. M., Friedl, M. A., Frolking, S., Wisser, D.: Global rain-fed, irrigated, and paddy croplands: A new high resolution map derived from remote sensing, crop inventories and climate data, International Journal of Applied Earth Observation and Geoinformation, 38, 321–334, doi:/10.1016/j.jag.2015.01.014 , 2015.
- Wagner, W., Hahn, S., Kidd, R., Melzer, T., Bartalis, Z., Hasenauer, S., Figa-Saldaña, J., de Rosnay, P., Jann, A., Schneider, S., Komma, J., Kubu, G., Brugger, K., Aubrecht, C., Züger, J., Gangkofner, U., Kienberger, S., Brocca, L., Wang, Y., Blöschl, G., Eitzinger, J., and Steinnocher, K.: The ASCAT soil moisture product: A review of its specifications, validation results, and emerging applications, Meteorol. Z., 22, 5–33, 2013